# Could an extreme cold Central European winter such as 1963 happen again despite climate change?

Sebastian Sippel[1], Clair Barnes[2], Camille Cadiou[3], Erich Fischer[4], Sarah Kew[5], Marlene Kretschmer[1,6], Sjoukje Philip[5], Theodore G. Shepherd[6, 7], Jitendra Singh[4], Robert Vautard[8], and Pascal Yiou[3]

[1]Leipzig Institute for Meteorology, Leipzig University, Stephanstr. 3, 04103 Leipzig, Germany
[2]Grantham Institute, Imperial College, London, SW7 2BU, UK
[3]Laboratoire des Sciences du Climat et de l'Environnement, UMR 8212 CEA-CNRS-UVSQ, IPSL and U Paris Saclay, 91191 Gif-sur-Yvette CEDEX, France
[4]Institute for Atmospheric and Climate Science, ETH Zurich, Universitätstrasse 16, 8092 Zurich, Switzerland
[5]Royal Netherlands Meteorological Institute (KNMI), De Bilt, The Netherlands
[6]Department of Meteorology, University of Reading, Reading, UK
[7]Jülich Supercomputing Centre, Forschungszentrum Jülich, Jülich, Germany
[8]Institut Pierre-Simon Laplace, CNRS, Université Paris-Saclay, Sorbonne Université, France

**Correspondence:** Sebastian Sippel (sebastian.sippel@uni-leipzig.de)

**Abstract.** Central European winters have warmed markedly since the mid-20th century. Yet cold winters are still associated with severe societal impacts on energy systems, infrastructure and public health. It is therefore crucial to anticipate storylines of worst-case cold winter conditions, and to understand whether an extremely cold winter, such as the coldest winter in the historical record of Germany in 1963 ($-6.3°C$ or $-3.4\sigma$ seasonal DJF temperature anomaly relative to 1981-2010), is still possible in a warming climate. Here, we first show based on multiple attribution methods that a winter of similar circulation conditions to 1963 would still lead to an extreme seasonal cold anomaly of about $-4.9$ to $-4.7°C$ (best estimates across methods) under present-day climate. This would rank as the second-coldest winter in the last 75 years. Second, we conceive storylines of worst-case cold winter conditions based on two independent rare event sampling methods (climate model boosting and empirical importance sampling): a winter as cold as 1963 is still physically possible in Central Europe today, albeit very unlikely. While cold winter hazards become less frequent and less intense in a warming climate overall, it remains crucial to anticipate the possibility of an extreme cold winter to avoid potential maladaptation and increased vulnerability.

# 1    Introduction

The winter of 1962/1963 (hereafter just referred to as winter 1963) was the coldest in the historical record in many Central European countries ('winter' defined as December-January-February temperature average). For example, in a record extending back to 1880, it fell below the second-coldest winter (1940) in Germany by a margin of -0.5°C (Deutscher Wetterdienst[1]). The extreme cold reached across large parts of mainland Europe, reaching from the Eastern and Northern Baltic sea regions to Western Europe (Hirschi and Sinha, 2007), with well-documented impacts on humans and ecosystems (Eichler, 1970). For instance, large European water bodies, such as the Ijsselmeer and Lake Constance, as well as rivers such as the Rhine and Rhone, and large parts of the Baltic sea, were frozen.

## Dynamical origin of winter 1963

Over the North Atlantic and Europe, a positive geopotential height anomaly prevailed over Iceland and western Scandinavia, and a negative anomaly stretched across the continent from the Atlantic coast off the Iberian Peninsula to Western Russia (Greatbatch et al., 2015). At the surface, positive pressure anomalies over Iceland and negative anomalies to the west of the Iberian Peninsula and across the Mediterranean are associated with a negative anomaly in the North Atlantic Oscillation (NAO) index. While the NAO was not extremely negative, as the low was displaced to the east of the Azores (Cadiou and Yiou, 2024), the jet stream and associated westerlies were split and displaced far to the north and south of their usual position (O'Connor, 1963). This atmospheric circulation anomaly favored easterly and north-easterly winds and advection of cold air into mainland Europe, a known synoptic situation that can lead to extreme cold air outbreaks over Central Europe (Loikith and Neelin, 2019). Notably, in winter 1963, the blocked anticyclonic conditions over North Western Europe persisted over most of the winter, inducing rather stationary, cold easterly winds. These cold conditions were further enhanced through the presence of snow by an extremely high albedo reflecting solar radiation. The long cold tail of the observed distribution in Fig. 1a due to winter 1963 is not an artefact of the short observed record, but can be well explained by the prevailing circulation patterns (Loikith and Neelin, 2019): under normal winter conditions Central Europe is under westerly flow, while the climatologically coldest air resides to the East. Rare, anomalous reversals towards easterly flow thus create the observed long, non-Gaussian cold tail (Loikith and Neelin, 2019), which is further enhanced by snow-albedo feedbacks (Groisman et al., 1994).

The European temperature pattern was embedded in Northern Hemisphere temperature anomalies with severe cold over the Eastern US, and very mild conditions over the Labrador Sea and Alaska (O'Connor, 1963; Hirschi and Sinha, 2007). These spatial patterns of anomalous temperatures across the Northern Hemisphere resulted from pronounced hemispheric atmospheric circulation anomalies (O'Connor, 1963), possibly related to variations in the Quasi-Biennial Oscillation and associated strong easterly winds in the equatorial troposphere (Greatbatch et al., 2015). In late January 1963, a sudden stratospheric warming event took place with the associated weakened polar vortex and may have helped to maintain persistence in the anomalous

---

[1]https://www.dwd.de/EN/ourservices/cdc/cdc.html, accessed 17.07.2023

conditions throughout February in Europe (Greatbatch et al., 2015). Persistent weak stratospheric polar vortex states are known to be associated with cold conditions over Europe (Kretschmer et al., 2018).

Overall, the winter of 1963 serves as a canonical illustration of an extreme cold Central European winter. If a European winter of similar intensity as in 1963 were to re-occur in today's world, it would almost certainly have severe societal implications: Extreme cold winter temperatures cause adverse impacts in many sectors such as health, transportation, infrastructure and energy (e.g., Pinto et al., 2024). As an example in the health sector, cold extremes are associated with increases in respiratory diseases affecting in particular the elderly and more vulnerable population (Curtis et al., 2017; Hajat and Haines, 2002), and

increased mortality due to cardiovascular diseases (Curtis et al., 2017; Charlton-Perez et al., 2021). Vulnerable groups such as the temporary workforce, outdoor workers or the lower-income population with insufficient or precarious shelters, and insufficient access to energy or fuel, may be also strongly affected (Pinto et al., 2024). As another example, the energy sector is particularly affected by cold extremes. On the demand-side, there is a close relationship of temperature with demand for heating (Petrick et al., 2010; Zeniewski et al., 2023). On the supply side, several past events such as Texas in February 2021 have caused

blackouts or near blackouts due to electric grid overload, or failure in sufficient power generation (Gruber et al., 2022). The combination of high demand and low supply may lead to associated price spikes from cold winter temperatures for electricity or natural gas, which may even lead to energy or fuel deprivation and mortality for vulnerable groups (Chirakijja et al., 2019). Given the vulnerability of different sectors to extreme cold temperatures in Europe and beyond, adaptation to those events, such as winterization of the energy sector (Zakeri et al., 2022) or cold wave preparedness and contingency measures in the

health sector (Pinto et al., 2024), is crucial. The adaptation question may be particularly important in the context of ongoing societal transformations in Europe, such as an ageing population and the transformation of the energy system with a larger share of renewables in many European countries. Despite the strongly warming European winter climate with a sequence of mild winters for more than a decade in Central Europe, extreme cold temperatures can still occur (Quesada et al., 2023)(Pinto et al., 2024). Hence, potential extreme cold temperatures, despite a warming average winter climate, must be considered in

adaptation strategies to avoid the risk of maladaptation.

    In this study, we use winter 1963 as a storyline of a 'worst-case' cold winter over Central Europe, and we assess two research questions, by combining multiple attribution methods: First, if a winter atmospheric circulation similar to 1963 were to re-occur in present-day climate, what would be the intensity in terms of cold temperatures? Second, is a winter as cold as 1963 or colder still possible in Central Europe today? The manuscript proceeds as follows: We describe the attribution methods used to address

both research questions (Section 2). Next, we present and discuss our results (Section 3), followed by conclusions (Section 4).

## 2 Methods and Data

### 2.1 Data processing and analysis choices

In this paper, we study potential 'worst-case' cold winter conditions in Central Europe. We focus our analysis as an illustration on the spatial domain of Germany[2]. Our analysis is based on the ERA5 reanalysis dataset (Hersbach et al., 2020), and we analyse daily and seasonal mean (DJF) temperature anomalies. Daily anomalies are computed by subtracting a 31-day moving average seasonal cycle based on the 1981-2010 reference period. Seasonal anomalies are calculated from daily anomalies. The analysis of warming trends across the paper is conducted in terms of global mean warming levels: That is, the trends are estimated as a function of the 4-year smoothed global mean surface temperature in ERA5 similar to World Weather Attribution analyses (Philip et al., 2020).

### 2.2 Dynamical adjustment of temperature trends

We start our analysis in Fig. 1 by characterising circulation-induced and residual (thermodynamical) trends of the domain-averaged winter temperature time series over the Germany domain. Dynamical adjustment is a technique in climate science, which aims to estimate the influence of atmospheric circulation on a target surface climate variable, such as surface air temperature (Wallace et al., 1995; Smoliak et al., 2015; Deser et al., 2016). This method involves breaking down temperature variability into two components: a thermodynamical component and a circulation-induced component. The circulation-induced component is identified using a proxy of atmospheric circulation such as the spatial patterns of sea level pressure or geopotential heights in a regression setup: the estimated circulation-induced component is then the component that can be explained by the circulation proxy (Smoliak et al., 2015; Sippel et al., 2019). The residual component, unexplained by the circulation proxy variable is then assumed to contain the thermodynamical component, which includes forced thermodynamical changes, and any unexplained variability. These approaches assume a linear separation between both components, which is a limitation of these techniques. In this study, we pursue two different approaches to dynamical adjustment. Both approaches estimate the circulation-induced component of daily mean winter temperature over the target region (Germany domain), using a regularized linear regression technique, called 'elasticnet regression' (Zou and Hastie, 2005). The methodological setup follows an earlier dynamical adjustment study (Sippel et al., 2019). Elasticnet regression is a statistical learning technique that uses a regularization scheme to handle the large number and collinearity of circulation predictor grid cells. The first dynamical adjustment approach (dark blue line in Fig. 1) uses ERA5 to train the regression model. Sea level pressure (SLP) grid cells are used as predictors, and cover a spatial domain of Europe along with the North Atlantic (45°W-35°E, 22°N-72°N). This approach is identical to Sippel et al. (2020). Because of the risk of overfitting when training based on the relatively short ERA5 period, we estimate for each year $y$, an individual regression model that does not use $y$ and the adjacent years $y-1$ and $y+1$.

We also use a second dynamical adjustment approach, based on a slightly different method (Singh et al., 2023), where we train the regression model based on the CESM2-LE (Rodgers et al., 2021). Geopotential height patterns at 500 hPa over a

---

[2]https://gadm.org/download_country_v3.html

Europe and North Atlantic domain (30°W-35°E, 16°N-80°N) are used as predictors. We subtract the domain-average mean trend of geopotential height patterns to account for the long-term column expansion due to warming. The obtained regression model is fully independent from the observational record (since trained on CESM2-LE), and is applied subsequently to the ERA5 dataset for comparison (light blue lines in Fig. 1). This approach has been used in a recent study on summer heat extremes (Singh et al., 2023). This second approach is only slightly less effective in explaining temperature variability (see Fig. 1), even though it is trained on a completely different dataset and on a different circulation proxy variable, and yields almost identical results in terms of circulation-induced vs. residual thermodynamical trends.

## 2.3 Circulation analogues for translating 1963-circulation to present-day climate

We continue our analysis by assessing how winter 1963 could look like in a present-day climate using a circulation-conditional 'thermodynamical translation' method in Fig. 2. The 'thermodynamical translation' method allows the translation of temperatures from one historical event to the current conditions. It follows the formalism of separation of dynamical and thermodynamical contributions described in an earlier attribution study (Vautard et al., 2016), which is relatively similar to Shepherd (2016). It assumes that, in a stationary climate, an atmospheric circulation induces an expected conditional temperature at each grid point and a given distribution around that average value. This distribution can be calculated in practice by selecting temperatures occurring in analogue flows.

Vautard et al. (2016) applied this concept with two climate states: a factual climate and a counterfactual climate, using the weather@home simulations of the wet winter of 2014, and concluded that about a third of the change in the monthly amount of the January temperature was due to circulation changes. As in Vautard et al. (2023), we use here analogues along the ERA5 reanalysis time series from 1950 to 2022 and directly regress the temperature trends across analogues as a function of their associated global temperature in order to define a scaling relation, which depends on the circulation. Circulation is characterized by the 500 hPa streamfunction over the [-10 to +40°E ; 40 to 70°N] domain. Analogues are characterized by their anomaly correlation coefficient (ACC) between streamfunction fields. Along the 1963 winter, for each day, we collect the best 1% winter analogues (64 days), which are apart from each other by at least six days. The conditional regression coefficient is then calculated for temperature at each grid point and day, it is multiplied by the difference in global temperatures between 1963 and 2022, and the result is added to the 1963 temperature to simulate daily temperatures in 2022. The global warming level is calculated from smoothed global temperatures from ERA5, using a moving centered 4-year average of the global temperature with available data, for reanalyses and models, accounting for series ends in ERA5.

## 2.4 CESM2 amplification of tail events to translate the 1963 winter to the present-day

In addition to the 'thermodynamical translation' method, it may be instructive to assess how the Central European winter temperature distribution changes as a function of global mean temperature change in a large climate model ensemble. Therefore, we assess how cold tail events change in the CESM2 large ensemble, using quantile regression (Koenker and Hallock, 2001). That is, we linearly estimate the value of the 0.84th percentile that corresponds to a '119-year event' (the best-estimate return time of winter 1963, see below) in the winter temperature distribution, with a co-variate on global mean temperature change

in the large ensemble. Next, we estimate how a percentile corresponding to a '119-year event' would have changed if we use the global mean temperature change from ERA5 as an input to our quantile regression model. This approach is conditional on a tail event, but not conditional on the atmospheric circulation. The CESM2 model is used as it is shown to perform well for European regional climate (Deser and Phillips, 2023), and indeed the distributions between CESM2-LE and ERA5 compare well visually (Fig.3a). The standard deviation of the detrended 1950-2023 winter seasonal temperature distribution is 1.40°C,
which is slightly smaller than the winter temperature standard deviation in ERA5 (1.68°C). We estimate intensity changes, but we do not derive probabilistic return period or frequency estimates from the model. The model shows only a very small ensemble mean circulation-induced trend over the historical period (Fig. 10c in Deser and Phillips (2023)).

## 2.5 Unconditional approach to translate 1963 to the present-day

To investigate the unconditional trends in observed winter temperatures (shown in Fig. 1 and Fig.3a), we follow the methods
outlined in the World Weather Attribution protocol (Philip et al., 2020). A nonstationary generalised extreme value distribution (GEV) is fitted to the ERA5 DJF mean temperatures (Coles, 2001), using maximum likelihood methods to estimate the parameters of the statistical model. We note that the DJF mean temperatures are not block minima and so the assumptions underpinning the use of the GEV are not, strictly speaking, met. However, comparison of the DJF mean time series with the annual minima of 90-day running mean temperatures shows that the two are highly correlated (Pearson correlation coefficient:
0.98), and the same analysis applied to the minimum of the 90-day running averages produced almost identical results. For consistency with the other analyses, results are therefore reported for DJF mean temperatures. All analysis was carried out using the extRemes package in R (Gilleland and Katz, 2016); model fitting was carried out over negative temperatures to ensure stability of the parameter estimates (Coles, 2001).

The distribution of the DJF mean temperatures is assumed to be dependent on the 4-year smoothed ERA5 Global surface
air temperature anomaly (GSAT) with respect to 1981-2010, with the location parameter assumed to vary linearly with the GSAT anomaly. Alternative nonstationary models, in which the scale or shape parameters were also allowed to vary with GSAT, were also fitted, and likelihood ratio tests used to select the most appropriate model to represent this time series (Coles, 2001) (Theorem 2.7); however, none of the more complex models were found to improve the fit enough to justify the use of an additional parameter. Having estimated the parameters of this nonstationary distribution, we compute the return period of
the observed 1963 DJF temperatures at the 2023 GSAT anomaly. A bootstrap procedure is used to estimate a 95% confidence interval for this return period (Efron and Tibshirani, 1994): whole years are resampled with replacement to construct a synthetic dataset of the same size as the original, and GEV parameters estimated from the resulting sample are used to compute the 2023 return period of the 1963 DJF temperatures. This procedure is repeated 1000 times, and the 0.025- and 0.975- quantiles of the estimated return periods give the bounds of the central 95% confidence interval for the return period of the 1963 event in the
2023 climate.

## 2.6 Empirical importance sampling to evaluate whether a worst-case cold winter such as 1963 is still possible

We use the approach of two recent studies (Yiou and Jézéquel, 2020; Cadiou and Yiou, 2024), which emulate temperature trajectories with a stochastic weather generator (SWG) based on analogs of atmospheric circulation. This SWG approach follows a Markov chain with hidden states of atmospheric circulation, using stochastic reshuffling of daily atmospheric fields to generate atmospherically-consistent alternative trajectories of climate events on a sub-annual time scale. For each day, the 20 best analogs of geopotential height at 500hPa (Z500) are determined by minimizing Euclidean distance over a circulation domain covering Europe and the North Atlantic (20°W to 30°E; 30°N to 70°N). The reshuffling process assigns weights favoring analogs with lower temperatures over Germany, using importance sampling techniques with weights for colder days to enhance the simulation process towards more extreme states. This targeted selection of colder analogs through importance sampling deviates from the uniform weighting approach typically associated with a standard emulator of temporal sequences, as delineated in Yiou (2014). The implementation of a calendar constraint further refines our simulation, ensuring that the seasonal cycle is accurately represented and aligns with climatological expectations. The calendar and importance sampling weights are chosen and combined as described in Cadiou and Yiou (2024). The reshuffling is performed on a daily basis, which allows for flexibility and the incorporation of daily variability. To emulate the winter of 1962-1963 in Germany (DJF), we conduct two sets of 1000 simulations using analogs from either a counterfactual period (1950-1999) or a factual period (1972-2021). Each simulation is initialized on December 1st, 1962, and the analog selection process is performed for a duration of 90 days. Analogs from the winter of 1962-1963 are excluded from the simulations to ensure the independence of the simulated events from the observed one.

## 2.7 Climate model boosting to evaluate whether a worst-case cold winter such as 1963 is still possible

Here, we first analyse a 30-member CESM2 initial condition large ensemble ('CESM2-ETH' from now onwards), which spans the time period from 2005 to 2035. The ensemble thus encompasses 900 winter seasons (DJF), and is bit-by-bit reproducible, which is a requirement for climate model boosting. The CESM2 simulations are forced with historical CMIP6 forcing from 2005 to 2014 and SSP3-7.0 from 2015 to 2035, following the same protocol as the CESM2 large ensemble (Rodgers et al., 2021). The simulations branch off from a transient historical simulation in 2005 via a round-off perturbation in the atmospheric initial conditions. The analysis of the 30-member ensemble, and the identification of some of the worst-case events, serves as the input to climate model boosting.

Second, we use climate model boosting as a technique that was developed and used in recent studies to generate physically plausible storylines of worst-case events (Gessner et al., 2021; Fischer et al., 2023). The idea of climate model boosting is to re-initialize a climate model about 5-20 days before an extreme event occurs with a round-off perturbation. This allows that different but physically plausible realizations of that particular event can be generated, similar to an ensemble weather forecast. Hence, the tail behaviour can be explored. Our boosting approach aims to generate winters with very cold seasonal temperatures. This process leads to perturbed events that are generated with varying lead times. Specifically, the perturbation is done on the specific humidity q at each grid point, and for each lead time, in the order of $10^{-13}$ to produce 50 ensemble

members per lead time. The perturbation was kept as small as possible to conserve mass, energy, and momentum up to the
precision of a round-off error. After the initial perturbation, the fully coupled model was run for approximately 60 days,
where the ensemble spread is small for 4-5 days and grows rapidly thereafter. The boosting analysis begins by identifying the
coldest December in the CESM2-ETH ensemble during the 2020s. Subsequently, this simulation is perturbed and re-initialized
for each day from 1st to 15th December, resulting in a total of 750 ensemble members (50 members per day). This initial
boosting step yields a relatively well-constrained ensemble for the first two to three weeks of December. Towards the end of
December, a few individual simulations remain notably cold even into January. To perform a second-order boosting, we select
the two coldest simulations from January among the first-order boosted simulations. The two coldest simulations were selected,
because computational resources allowed it. These selected simulations are then re-initialized for each day between 1st to 15th
January, resulting in 50 ensemble members per initialization day and a total of 1500 simulations. A recent independent paper
followed a similar strategy to re-initialize CESM1.2 in order to generate events with high return periods (Ragone and Bouchet,
2021).

## 3 Results and Discussion

### 3.1 Central European winter temperatures in a warming climate

Over Central Europe, using the Germany domain for illustration, winter temperatures have increased by about 2.5°C since the
mid-20th century (2014-2023 w.r.t 1951-1980 in ERA5, Fig. 1a). The circulation-induced component of temperature variabil-
ity is separated from a residual component that is left unexplained by circulation and expected to contain thermodynamical
components, following the Methods section (Fig. 1). Circulation-induced variability shows a Pearson correlation of $R = 0.95$
with the observed, detrended time series of the DJF temperature anomaly over Germany, thus confirming that atmospheric
circulation is the dominant driver of inter-annual winter temperature variability (Fig. 1a). The residuals show a positive trend,
which cannot be explained by the atmospheric circulation, and likely reflects the direct thermodynamical effect of warming
(Fig. 1b). The warming is consistent with a trend towards less frequent and intense very cold days and nights at the global
scale (Seneviratne et al., 2021). The direct thermodynamical warming is expected to be further amplified by feedbacks such as
snow-albedo effects and because of the weakening of cold air advection in the mid-latitudes due to Arctic amplification, which
acts to also reduce mid-latitude wintertime temperature variance (Screen, 2014; Schneider et al., 2015; Blackport and Kushner,
2016; Holmes et al., 2016; Gross et al., 2020; Tamarin-Brodsky et al., 2020; Blackport et al., 2021).

Beyond thermodynamical effects, however, atmospheric circulation changes have contributed to warming winter tempera-
tures over Central Europe, as evidenced by the circulation trend in Fig. 1a. The positive circulation contribution to warming,
with less frequent cold spells, is consistent with other studies (Vautard and Yiou, 2009; Faranda et al., 2023), and reflects more
frequent zonal flows and less frequent blocked flows in the recent past (Blackport and Fyfe, 2022; Faranda et al., 2023). Model
simulations, including CESM2, however, show little or no evidence for pronounced forced changes (Blackport and Fyfe, 2022),
and mechanistic links between Arctic amplification, sea ice loss and mid-latitude circulation remain uncertain (Blackport and
Kushner, 2016; Outten et al., 2022). Therefore, the future of forced regional atmospheric circulation changes remains highly

uncertain (Shepherd, 2014; Zappa and Shepherd, 2017; Blackport and Fyfe, 2022; Faranda et al., 2023). It is beyond the scope of our study to investigate models' circulation trends, which have been analysed in several past studies (Blackport and Fyfe, 2022; Faranda et al., 2023). However, the discrepancy between model simulations and observations carries important implica-

tions for understanding and constraining the potential for future cold winters: If the circulation trend was indeed forced, but missed by climate models, it would be less likely to see winter 1963 circulation conditions again. If, however, the dynamical trend would be due to natural variability, it may revert and circulation conditions similar to 1963 may appear again with equal probability. It is the latter scenario that would bring severe risks for European societies. Therefore, we will focus on a storyline approach, which has been conceived precisely for the conditions of large dynamical uncertainties (Shepherd, 2016). In this

context of dynamical uncertainty, we analyse in the next subsection how winter 1963 circulation conditions would play out in a present-day climate.

### 3.2    What winter 1963 would look like today

To address our first research question, we use the method of circulation analogues to translate a hypothetical recurring winter 1963-type circulation into present-day climate (see Methods). The analogue procedure is useful here, because while winter

1963 on a daily basis was rather persistently cold, there are several other years that show cold spells on shorter time scales (Fig. 2a). The thermodynamical translation reveals that winter 1963 would be about 1.4°C warmer in a present-day climate in Germany (Fig. 2b-d). A south-west to north-east contrast in the warming may be explained as the cold anomalies arise from advection from the coldest climatological regions, and those coldest climatological regions are where the thermodynamical warming is the greatest. This approach can be compared to a conditional event attribution (Trenberth et al., 2015; Shepherd,

2016; van Garderen et al., 2021), where the role of atmospheric circulation is held fixed and we test how the thermodynamical component would have plausibly changed between 1963 and today's climate. Hence, these values are conditional on the 1963 atmospheric circulation and do not consider the changes in likelihood of such atmospheric circulation, or sequences of circulations. Unlike in most attribution studies, however, the event does not occur today but in a past period, with a projection to today's climate.

Next, we compare the results to a different approach based on an analysis of large ensemble (LE) simulations with the CESM2 climate model (Danabasoglu et al., 2020; Rodgers et al., 2021). CESM2-LE has been shown to capture the historical climate evolution in Europe (Deser and Phillips, 2023), with ensemble members that show accelerated warming also indicating a similar sea level pressure evolution to observations (Deser and Phillips, 2023). Despite the overall warming trend, the large ensemble suggests that deviations of more than 5°C below normal values can still occur and will continue to be possible for

several decades, albeit less frequently (Fig. 3a, grey lines). A quantile regression on the percentile corresponding to a 119-year event (which corresponds to the best-estimate return period of winter 1963 based on statistical analysis elaborated below) of cold winter temperatures on global mean temperatures in CESM2 reveals a regression coefficient of 1.6°C per degree of global warming. Hence, cold winter events with a 119-year return period over Germany amplify by 1.6°C per degree of global mean warming. With the ERA5 global mean warming of 1.0°C since 1963, this would imply an increase of about 1.6°C for cold

winter events with a 119-year return level (Fig. 3b). This approach implicitly conditions on tail events, but not on a specific type

of atmospheric circulation. This estimate is independent to the circulation analogue approach, but the estimated warming of +1.6°C of such a tail event is very similar to +1.4°C to the circulation analogue approach. In contrast to the cold tail, the mean of the winter temperature distribution only changes by 1.2°C per degree of global mean warming in CESM2, thus comparable to the mean thermodynamic trend of 1.0 to 1.1°C revealed by dynamical adjustment. We acknowledge that more extreme cold events than the '119-year event' studied above would potentially show even a higher amplification (Tamarin-Brodsky et al., 2020). This is because a weaker meridional temperature gradient, and large warming over Arctic regions (Pithan and Mauritsen, 2014) implies particularly strong warming of cold extremes in Central Europe that are caused partly by advection from those regions (Tamarin-Brodsky et al., 2020).

For comparison, we now turn to a statistical analysis of observations as used in World Weather Attribution studies (Philip et al., 2020), which uses a covariate on global mean temperature and is not conditional on the atmospheric circulation. We use a fit based on a longer period into the past, 1900-2023, using the DWD time series for the Germany domain, and with the global temperature covariate taken from the ERA-20C reanalysis (Poli et al., 2016). This statistical analysis indicates a best-estimate increase of 1.6°C since 1963 with an uncertainty range from 0.9 to 2.2°C (Fig. 3b). The 1963 winter event has thus been a very rare event with a return period of 119 years in 1963 (95% confidence interval 46 to 1102 years), and would be even less likely today due to the warming (371 years in 2021, with uncertainty 97 up to 7680 years). A statistical fit based on the shorter (yet still quite long) post-1950 period would yield a larger estimate of winter temperature change between 1963 and today (best estimate of +2.0°C, rather than +1.6°C when based on the longer period). This is likely because the post-1950 positive circulation-induced trend is congruent with thermodynamical warming, and the short period estimate would thus implicitly assume that the total winter temperature trend over Germany, including the circulation effects, is forced. Hence, the circulation trends would implicitly be assumed as forced effects through the global temperature covariate. This illustrates the potential pitfalls of using relatively short time series in an unconditional manner. It also shows that the storyline approach is not predisposed to exaggerating the effects of climate change compared to the probabilistic approach, as has sometimes been claimed (see García-Portela and Maraun (2023)). Overall, all attribution approaches employed here, including conditional and unconditional methods, agree that an extreme winter cold anomaly such as in 1963 would be warmer in today's climate with estimates of about +1.4 to 1.6°C, yet would still be very cold (Fig. 3b). A hypothetically recurring 1963-like extreme event in today's climate (i.e., similar circulation or similarly low likelihood) would thus still lead to plausible winter seasonal cold anomalies of -4.7 to -4.9°C (best estimates across the three methods), with a maximum uncertainty range of -5.4 to -4.1°C. Such a temperature anomaly would still be the second-coldest winter in Germany in the last 75 years, second only to the original 1963. The attribution statements are summarized in Table 1. Our analysis thus shows that while cold winters have become less frequent, a very cold winter remains a possibility. Potential mal-adaptation to the very mild winters experienced in the last decade would thus imply high societal risk.

## 3.3   Are 1963 temperatures still possible today?

We now explore our second research question, namely whether a winter as cold as 1963 would still be possible in today's climate. Based on the statistical GEV analysis presented in Subsection 3.2, cold temperatures such as in winter 1963 would still

be possible today, albeit very unlikely (best estimate of a 371-year return event). However, the statistical analysis is based on the relatively short observational record, with large uncertainties, and here our goal is to develop storylines of such cold winter temperatures based on independent rare event sampling methods. A recent study has shown that winter temperatures as cold as 1963 in France are still possible but very unlikely today (Cadiou and Yiou, 2024). This finding was based on a statistical resampling algorithm known as empirical importance sampling (Yiou and Jézéquel, 2020), which is designed to sample distributional properties of rare events. The rare event sampling is achieved by resampling circulation analogues of recent winters that lead (if combined persistently) to very cold conditions. Here, we compare this methodology to an independent approach to assess the physical plausibility of very rare events: Climate model boosting (Gessner et al., 2021) generates physically plausible realizations of a particular extreme event. For example, climate model boosting has been used to show that the record-shattering Pacific North-West heatwave of 2021 could have been anticipated based on climate model simulations (Fischer et al., 2023). To ensure robustness of CESM2 boosting simulations, we first evaluate and compare winter temperature simulations from the 30-member CESM2-ETH ensemble that covers the 2006-2035 time period. We assess how circulation and albedo relate to daily and seasonally cold temperatures in CESM2 in comparison to ERA5. Circulation explains a large amount of daily (Fig. 4a) and seasonal (Fig. 4c) variability in both CESM2-ETH (Pearson correlation of $R_{daily}$=0.87 and $R_{seasonal}$=0.91, respectively) and ERA5 ($R_{daily}$=0.88 and $R_{seasonal}$=0.95). The circulation-induced temperature estimate is obtained via the dynamical adjustment method described above. Although most of the temperature variability is explained by atmospheric circulation, an unexplained residual remains for the coldest days (Fig. 4a) and seasons (Fig. 4c). In particular, the 1963 winter temperatures in ERA5, as well as the coldest winter in CESM2-ETH are underpredicted by circulation alone (Fig. 4a, c). It is well known that snow cover changes the albedo and thus the surface energy balance, leading to local cooling (Groisman et al., 1994). We find that daily albedo values are negatively correlated with temperature anomalies in both ERA5 and CESM2-ETH. We thus include surface albedo, as a proxy for changes in the surface energy balance, in the regularized linear model. With that addition, winter temperatures can be explained almost perfectly in both ERA5 and CESM2-ETH (Fig. 4d; R=0.96 and R=0.95, respectively), including the coldest observed and simulated winters. Moreover, the spatial anomaly patterns of temperature, circulation and albedo that combine to the coldest observed and two coldest simulated winter temperatures are remarkably similar between winter 1963 and the coldest winter simulations in CESM2 (Fig. 5, three upper rows). These winters show a strongly negative geopotential height anomaly over Western and Central Europe, and a positive anomaly over Iceland and Greenland. This situation induces cold temperatures through advection from the North-East, and is consistent with the discussion above on the dynamical origin of the 1963 extreme cold winter. Similar to 1963, the two coldest simulated winters show a positive albedo anomaly over Central and Eastern-Central Europe. We thus conclude that atmospheric circulation variability and surface albedo are the key proximate drivers for seasonal-scale cold winter temperatures over Germany and Central Europe, and these drivers and relationships appear to be realistically simulated by CESM2.

Next, we evaluate model boosting simulations for a hypothetical extremely cold winter of the 2020's in CESM2. Our boosting strategy is to generate seasonally-cold winters, which do not necessarily include record-breaking short-term cold spells. We reinitialize the coldest December in the 2020's in the CESM2-ETH ensemble (details in Methods). This 1st-order boosting results in a relatively well-constrained ensemble for the first 2-3 weeks in December with cold conditions (light blue lines rep-

335 resent 1st order boosting trajectories in Fig. 6a), with a reversal and large ensemble spread towards the climatology at the end of December. However, several individual simulations remain very cold well into January. For a 2nd-order boosting, we select the two coldest January simulations amongst the 1st-order boosted simulations, and re-initialize those, shown as dark blue trajectories in Fig. 6). Two hypothetical very cold 'worst-case' winters emerge from these simulations, which would produce seasonal temperature anomalies around -9 to -10°C (Fig. 6b), thus colder than the observed 1963 winter. Despite the intensity

of the cold conditions, the spatial patterns of circulation and albedo that produce these events are remarkably similar but even more pronounced than in 1963 (Fig. 5, 4th row). It is particularly remarkable that the model is capable of simulating such cold conditions in the 2020's, as the cold tail of the winter temperature distribution tends to warm faster than average conditions due to well-understood physical reasons (Tamarin-Brodsky et al., 2020). This is consistent with the analysis of the CESM2 winter amplification in Fig. 3b. For a comparison to model boosting, the empirical importance sampling follows the strategy

outlined in a recent study (Cadiou and Yiou, 2024). Because this approach is based on a statistical resampling of the historical record, and can only be applied to a relatively long 'recent' period (1972-2021), it is not directly comparable to the climate model boosting results. Nonetheless, the coldest winter indeed shows very similar spatial patterns of temperature anomalies and geopotential height anomalies (Fig. 5, 5th row). It also shows negative albedo anomalies, although the resampling strategy was not constrained by albedo. A few winters among the 1000 realizations of empirical importance sampling also remain

colder than the original 1963 (Fig. 6b).

Despite the close resemblance of the extreme cold simulated winters and the observed extreme winter of 1963, both rare event sampling methods have limitations that should be discussed. Empirical importance sampling only re-samples from the observed distribution of daily weather, and can thus create new events only through a recombination of past events. It is thus limited by the length of the available record, and cannot create unseen weather at a daily time scale. Moreover, the method

is based on the link between circulation and temperatures and disregards other drivers (such as surface albedo), which might explain its slightly less extreme temperatures. On the other hand, climate model boosting is based on the CESM2 climate model, and thus biases (beyond the mean) in the physical simulation of cold winter temperature anomalies would affect the simulated rare events. Moreover, the distribution of 'boosted' events depends on the event selection - and it may be possible that even colder conditions could be generated if other atmospheric initial conditions would be re-initalized (Gessner et al., 2021;

Fischer et al., 2023). Finally, it is not straightforward to obtain a return period estimate for the boosted events. Nonetheless, both independent rare event sampling methods agree that a winter season as cold as 1963 in Germany cannot be ruled out in a present-day climate, even though its occurrence remains very unlikely.

## 4 Conclusions

In this paper, we addressed two research questions that closely relate to the understanding and attribution of cold winter events

in a changing climate: First, we showed based on different attribution methods that a hypothetical 1963-like winter circulation anomaly today would result in a temperature anomaly of about -4.7 to -4.9°C seasonally relative to 1981-2010, which would be about 1.4 to 1.6°C warmer than the original 1963 winter in Germany. Three different conditional and unconditional attribution

methods' best estimates all fall within that range. This hypothetical 1963-like winter in today's climate would still represent the second-coldest winter of the last 75 years in Germany. In the post-1950 period, atmospheric circulation change, that is a more zonal flow and less atmospheric blocking, has contributed to accelerated warming over recent decades (Faranda et al., 2023; Blackport and Fyfe, 2022). However, it remains unclear whether this circulation-induced warming signal is forced or unforced (Faranda et al., 2023). Models project little forced future circulation change (Blackport and Fyfe, 2022), and the IPCC AR6 WGI SPM (2021) (Masson-Delmotte et al., 2021) concludes '...there is low confidence in projected changes in the North Atlantic storm tracks'. Hence, it cannot be excluded that a reversal in atmospheric circulation change, if the historical circulation trends were unforced, could lead to 1963-like winter circulation and hence very cold temperatures.

Second, we combined two recently developed independent rare event sampling methods and showed that temperatures as cold or colder than 1963 are still physically possible in a present-day and near-future climate. Such conditions are worst-case scenarios and thus unlikely, but cold extremes will continue to occur in a warming climate (Cattiaux et al., 2010; Quesada et al., 2023). Hence, if such a worst-case cold winter event would occur, it poses significant risks to human health, infrastructure, the energy system, and agriculture. Therefore, it is of vital societal interest to avoid a potential premature mal-adaptation to the exceptionally mild winter conditions of the past decade in Central Europe.

*Code and data availability.* All code and geoscientific data to reproduce this analysis will be made available upon publication in public archives.

*Author contributions.* All authors contributed to the idea and scope of the paper. S.S. performed the analyses with contributions from C.B., C.C., J.S., R.V., and P.Y. All authors discussed the results and S.S. wrote the manuscript with contributions from all authors.

*Competing interests.* One of the (co-)authors is a member of the editorial board of Weather and Climate Dynamics.

*Acknowledgements.* We thank Urs Beyerle for setting up the CESM2 climate model boosting experiments. This work received support from the European Union's Horizon 2020 research and innovation programme under grant agreement No. 101003469 ('eXtreme events: Artificial Intelligence for Detection and Attribution', XAIDA). We acknowledge the CESM2 Large Ensemble Community Project and supercomputing resources provided by the IBS Center for Climate Physics in South Korea, details provided in Rodgers et al. (2021), doi:10.5194/esd-12-1393-2021. Furthermore, we acknowledge 'Copernicus Climate Change Service (C3S) (2017): ERA5: Fifth generation of ECMWF atmospheric reanalyses of the global climate. Copernicus Climate Change Service Climate Data Store (CDS), 17.07.2023' and ECMWF for providing ERA5 and ERA-20C data.

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

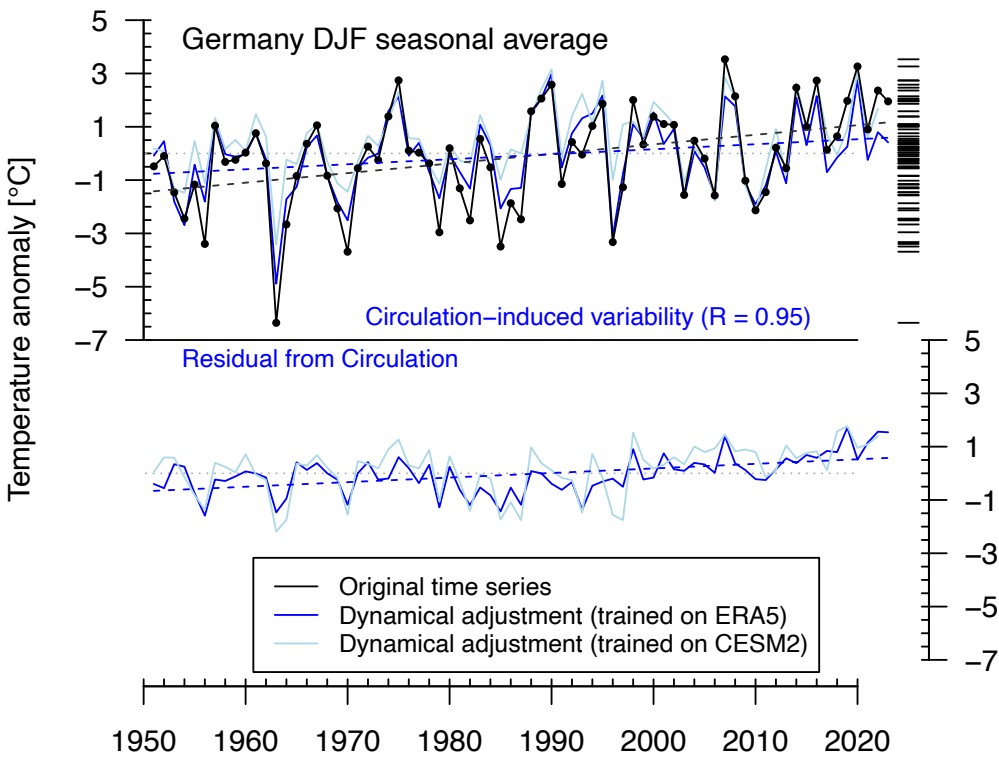

**Figure 1. Winter temperature anomaly time series over Central Europe (Germany domain) and long-term trends.** (top) 1951-2023 winter (DJF) temperature anomalies and the contribution of atmospheric circulation (blue line), and 'unconditional' long-term trend as a function of smoothed global mean temperature (black dashed). (bottom) Residual temperature anomaly time series when atmospheric circulation contributions are removed, and the trend of this 'circulation conditional' residual. Dashed lines show linear trends in the original time series (black), and the circulation-induced and residual component (blue).

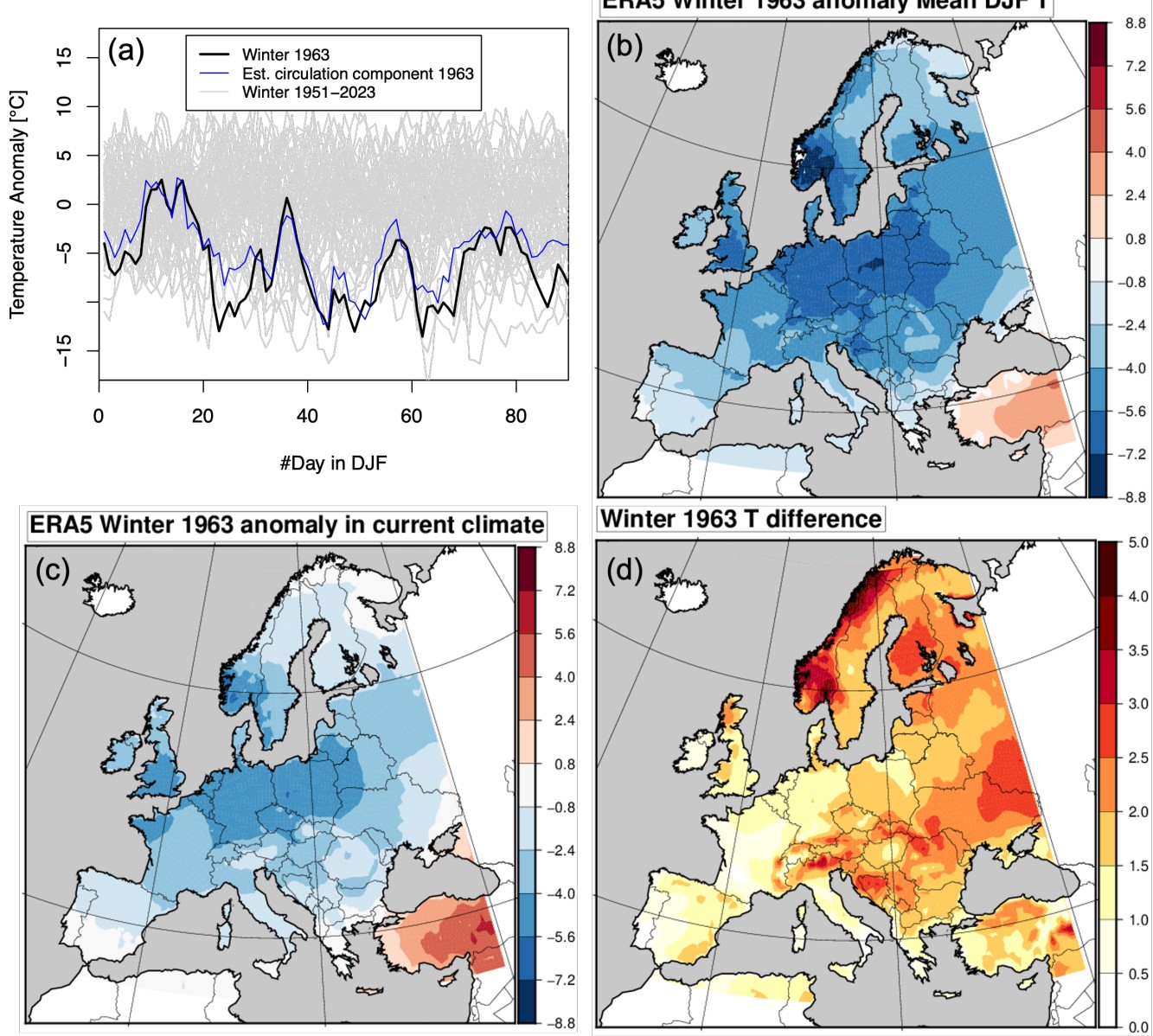

**Figure 2. Winter 1962/63 over Germany (-6.3°C DJF anomaly, -3.4σ), and its translation to present-day climate based on the 'analogues shift' method.** (a) Sub-seasonal 1962/63 winter temperature anomalies relative to the 1981-2010 baseline, with a sequence of cold waves in winter 1962/63 over Europe that are driven by atmospheric circulation anomalies (blue line, estimated from dynamical adjustment). (b) Spatial temperature anomalies in winter 1962/63 over Germany, (c) spatial anomalies for a hypothetical 1963 winter circulation under present-day climate using the 'thermodynamical translation' method based on circulation analogues (Vautard et al., 2023), and (d) the estimated temperature difference between (c) and (b), which may be interpreted as the thermodynamical change conditional on the 1962/63 atmospheric circulation.

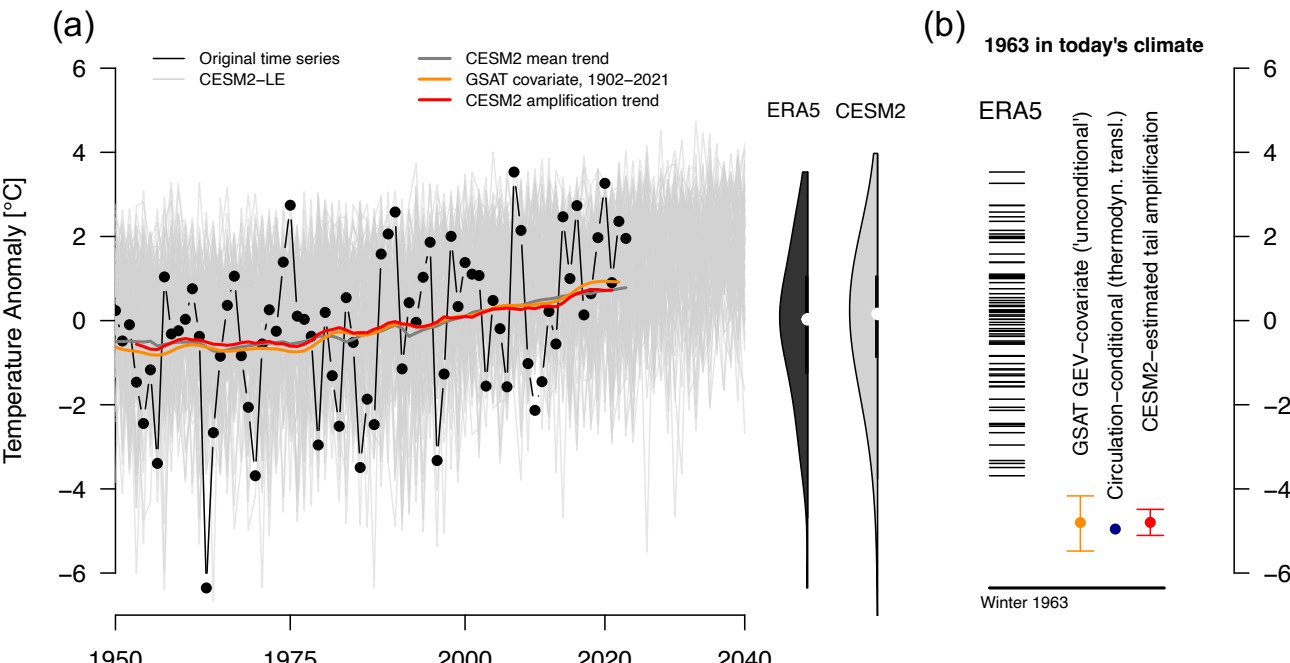

**Figure 3. Recasting the historical winter 1962/63 in a present-day climate.** (a) Observed Central European winter temperature anomalies in the context of a climate model large ensemble (CESM2-LE) and the long cold tail of the winter temperature probability distribution in ERA5 and CESM2-LE (shown at the right hand side of panel a). (b) Different approaches to translate winter temperatures of 1962/63 into present-day climate reveal warming of 1.4 up to 1.6°C (best estimates). Uncertainty bars reflect the parametric uncertainties of the GEV approach and the quantile regression approach for tail amplification in CESM2.

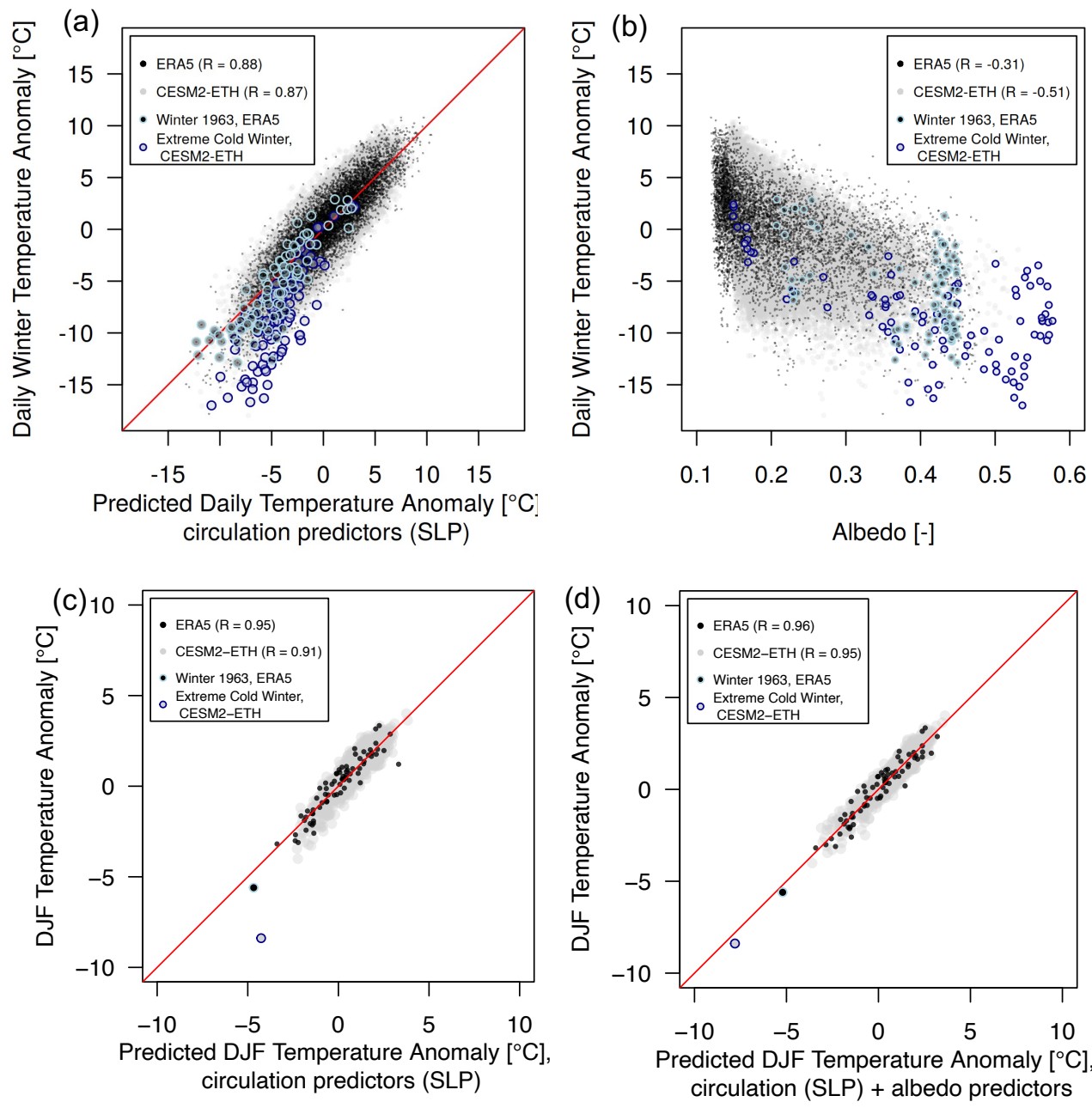

**Figure 4. Evaluation and comparison of CESM2-ETH model ensemble with observations.** (a,b)) Relationship of daily temperature anomalies with (a) a temperature estimate based on atmospheric circulation and (b) domain-average albedo in CESM2-ETH and ERA5. (c,d) Explaining seasonal temperature anomalies (c) with circulation only, and (d) with circulation and domain-average albedo. Albedo is needed as a predictor to explain the extreme cold tail in both ERA5 and CESM2-ETH.

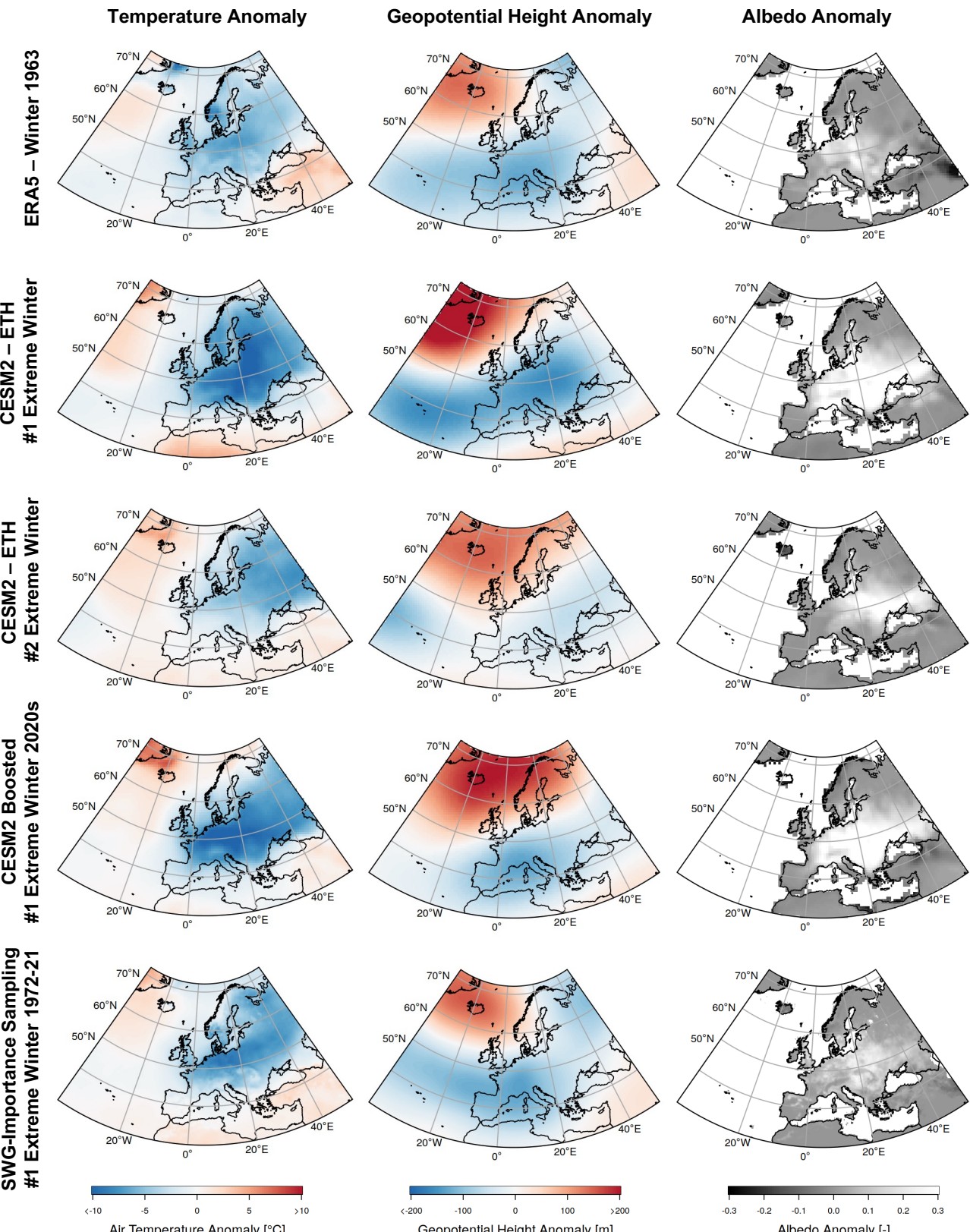

**Figure 5. Temperature, circulation and albedo anomalies for winter 1962/63 over Europe and CESM2-based cold winter storylines and empirical importance sampling.** Spatial anomaly patterns of (left column) near-surface air temperature, (middle column) geopotential heights at the 500 hPa level, and (right column) albedo for the European 1962/63 winter, the two coldest winters in the CESM2-ETH ensemble (#1 also highlighted in Fig. 4), the coldest boosted winter in CESM2, and the coldest winter generated by SWG-empirical importance sampling.

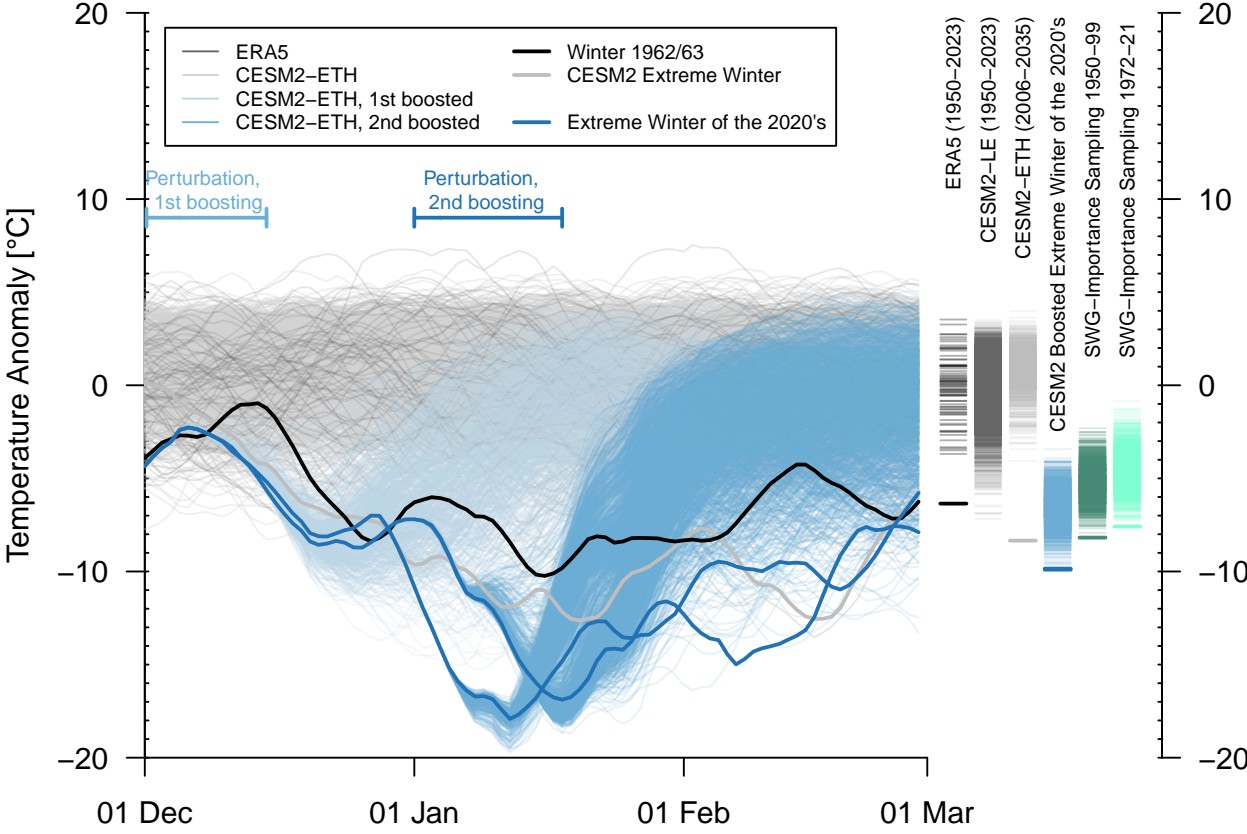

**Figure 6. Winter temperatures colder than in 1962/63 are unlikely, but still possible over Central Europe in present-day climate.**
(a,b) Storylines of hypothetical extremely cold winter temperatures over Central Europe. (a) Extreme cold winter of the 2020's in CESM2 obtained through 'model boosting'. First model boosting starting dates lie in the period December 1st to 15th (light blue ensemble), and second model boosting starting dates lie in the period January 1st to 15th (dark blue ensemble). (b) Probability distributions of ERA5 (1950-2023), CESM2-LE (1950-2023) and CESM2-ETH (2006-2035) are compared to worst-case winter storylines generated by CESM2 model boosting and stochastic weather generator empirical importance sampling based on ERA5. Both storyline methods indicate that extremely cold winter temperatures are still physically possible in the 2020's.

**Table 1.** Summary of potential attribution statements using conditional and unconditional attribution methods for a hypothetical '1963-like' winter in the 2020's.

| | |
|---|---|
| **Conditional on atmospheric circulation (using observed data only from 1950 onwards)** | If it occurred now, a winter with 1963 circulation conditions would be 1.4°C degrees warmer than in 1963 itself, but still 4.9°C colder than the average over 1981-2010. |
| | Average winter temperatures since mid-century have warmed by 2.5°C (2014-2023 w.r.t 1951-1980). A long-term change of 1.1°C since mid-century resulted from changes in circulation conditions which is congruent with global warming. It is not currently known whether this long-term change in circulation conditions is forced or a manifestation of internal variability. As a result, if the circulation change were to reverse, it is possible that the likelihood of extreme cold winters could increase over the next decade, in spite of global warming. |
| **Not conditional on atmospheric circulation (using observed data from 1901 onwards)** | If it occurred now, a winter with temperatures as rare as was the case in 1963 (a 1-in-119 year event) would be 1.6°C degrees warmer than in 1963 itself, but still 4.7°C degrees colder than the average over 1981-2010. |
| | A winter with temperatures as extreme as in 1963 (6.3°C below the 1981-2010 average) is three times less likely to occur (a 1-in-371 rather than a 1-in-119 year event). The likelihood of such a winter is expected to continue to decrease, in line with global warming. |
| **Not conditional on atmospheric circulation (using model data)** | Based on the CESM2 climate model large ensemble, if a winter with temperatures as rare as the observed 1963 winter occurred, it would be 1.6°C degrees warmer than in 1963. |