# Peer review of "Could an extreme cold Central European winter such as 1963 happen again despite climate change?"

_EGUsphere, 2023_

## Author Comment (AC1)

**Reply to Anonymous Referee #1 comment on egusphere-2023-2523,**

**doi:10.5194/egusphere-2023-2523 (Sippel et al.)**

In the following the reviewer comments appear in black with the author responses in blue.

This study has investigated the likelihood of a re-occurrence of the extremely cold European
winter of 1963 under present climate conditions and what such a winter would look like. To
investigate this, the authors employ a range of techniques, some involve accounting for the
dynamical features which gave rise to the extremely cold winter, while others are standard
statistical methods, e.g. extreme value analysis. All the methods gave approximately similar
answers, that such an event could occur today but is less likely, and if it did occur, the
temperatures would be approximately 1.5 degrees warmer than the original 1963 event.

The paper provides a clear and thorough assessment of the theoretical occurrence of the 1963
winter under today's climate. It is an interesting study, and would have been more interesting if
they had found that the winter could not occur today. However, since the conclusion is that an
extremely cold winter from the past is less likely to occur and would be warmer if it did occur
under today's warmer climate is not especially groundbreaking. The paper does raise the issue
of mal-adaption by society towards warmer winters, and I think this is very valuable. I also think
the comparison of the methods will be of interest to the community. Overall, I would
recommend the paper be published with minor revisions.

We thank the reviewer for the positive evaluation of our study, and we provide a more detailed
response to the issues raised below.

General Issues

1.  While the result that the cold winter can still occur and would be warmer is not wholly
surprising, the paper does makes a good point that society may be adapting under the
assumption of warmer winters while extremely cold winter are still very possible. I think
this could be raised up in the paper. Perhaps introducing the idea as part of the
motivation for the work so it is in the reader's minds as they go through.

Important point. We will provide a more solid case for a risk of mal-adaptation in the
Introduction.

2.  It is a potentially provocative finding that the unconditional statistical method of using a
fitted GEV gave as good a result as methods which incorporated knowledge of the
dynamics. Many statistical based studies are criticized for not including knowledge of the
dynamics of a situation, and here we have a case where this knowledge provided no
additional benefit, and for an extreme event no less. Obviously, this is a single case and it
could be random chance that the statistical method did so well. Given that the authors
have used so many different methods, I think it would be a nice addition to the paper for
the authors to briefly comment on how the methods compare, especially the
unconditional statistical method with the methods incorporating knowledge of the
dynamics.

There is indeed an important debate that discusses different conditional and unconditional
approaches to event attribution. We agree with the reviewer that the overall similar results may
occur by chance. But it may be noted nonetheless: The statistical method is fitted based on a long
1900-2023 period, with thus probably a relatively small dynamical trend over this long time
scale (hence the thermodynamical part likely dominates also the unconditional analysis, which may partly explain the overall similar results). Interestingly, if the statistical method would be
fitted on a shorter 1950-2023 period, the warming estimated obtained for a 1963-like winter
would be much larger, because of the dynamical trend on this decadal time scale. We will explain
this in more detail in the revised manuscript.

3.  The title is a little sensationalist and I'd suggest changing it to match the style of the WCD
journal.

Thank you for this feedback. We will consider changing the title in a revised manuscript.

4.  The second paragraph in the Results section discusses the failure of models to show
pronounced forced changes in atmospheric circulation. This is not a topic that is really
investigated by this study. CESM2 simulations were run to perform the model boosting
analysis, but there is no assessment of how or why or to what extent CESM2 fails to show
forced changes. The discussion reads more like a commentary on the failure of models in
this particular aspect, and does feel connected to the rest of the study. If this is a main
motivation or theme in this study, then this failure of models should be introduced in the
introduction and its implications on the use of model data in this study needs to be
assessed, not merely commented on. Without expanding on the issue raised in this
paragraph and incorporating it more fully into the study, I'd recommend removing the
paragraph, more specifically, sentences from line 204 to 212.

Thank you for highlighting this important point. Indeed it is not the goal of this study to
investigate the ability or failure of models with respect to simulating multi-decadal trends in
atmospheric circulation. There are many other studies that look at this phenomenon, and we cite
them (Blackport and Fyfe 2022, Faranda et al., 2023). But we also stress that it is currently not
understood whether the observed circulation-induced trend is indeed forced or not. For this
precise reason (uncertainty in forced dynamical changes), the conditional storyline approach
was conceived (Shepherd ). In the context of our study, CESM2 shows only a very small
ensemble mean circulation-induced trend over the historical period (Deser et al., 2023, Fig. 10c).
Hence, we use CESM2 to explore the worst-case in the sense that the observed circulation-
induced warming trend would be unforced, which would imply that very cold atmospheric
circulation patterns, such as 1963, may still recur. We will make this point (and its implications)
more clear in the revised paper.

5.  Figure captions for figures 2, 3, and6 need to be expanded to better explain what is being
shown in the figures. This is especially true of figures 3 which is important for the
paper.

We will expand substantially on the clarity and content of figure captions in a revised
manuscript.

Minor Issues
24: You use the expression "led to". This suggests some precession in time, that the pressure
anomalies occurred and then afterwards, there was a negative NAO anomaly. Perhaps change to
"resulted in" or "comprised".

Correct, we did not mean precession in time, but indeed "resulted in" or "associated with".

65-90: The dynamical adjustment method. Is it reasonable to separate the dynamic and
thermodynamic influences on surface temperature in such a linear way? Consider adding a
sentence or two discussing the caveats or limitations of this assumption/approach.

Important point. In the dynamical adjustment literature, this separation in both component
appears quite standard (Smoliak et al., 2015, Deser et al., 2016, Deser et al., 2023). But of course
it needs to be acknowledged that this is an assumption that may bear some caveats for the
interpretation.

78: "The first dynamical adjustment approach (dark blue line in Fig. 1) uses ERA5 to train the
regression model, and the spatial pattern of sea level pressure (SLP) over a circulation domain
over Europe and the North Atlantic." Possibly it is just my reading of the sentence, but it feels a
little awkward. The 'and' feels like it is part of the regression statement. Consider changing this
to "along with".

82: Start a new paragraph at "We use a second method…" It provides a cleaning break when
reading. Possibly change to "We also use a second…"
169-183: Please revise the structure of this paragraph. It jumps straight into what the method
leads to and only describes the method itself towards the end of the paragraph. I suspect this is
one of those cases where the author is so familiar with the method, he forgets that the reader
may not understand what he is talking about from the beginning.

OK, thanks for highlighting these (all 3 comments above). We will clarify the text in a revision.

174-181: Please explain in the paragraph why you use the single coldest winter in December for
the first boosting, but two coldest Januarys for the second boosting.

Our aim was to test (in the context of a "storyline") whether a winter of the 2020's could be even
as cold as the observed 1963 winter. Therefore we started with the 1st to 15th December of the
coldest December in the 2020's (there was a colder December in the 2000's which we did not
boost). We then realized that most ensemble members of this coldest December in the 2020's
were going back to the climatologically expected "normal" range in the following January, but
there were two ensemble members that showed exceptionally cold (but almost identically cold)
conditions in the following January. Because resources allowed it, we decided to boost both of
these events, which yield relatively similar "coldest DJF of the 2020's" conditions.

176: Please write the dates out in full. Using "01.12 to 15.12" could be written as "1st to 15th
December".

180: Again, write the dates out in full.

219: Remove the double brackets.

OK, thanks (to all 3 above).

245: The return period of the 1963 event was 119 years. For such an event to occur today, the
return period would be 371 years. However, the uncertainty for the occurrence today is 97 to
7680 years. That is to say, the return period of the event today may still be 119 years at the 95%
level. This should be commented on.

We agree with the reviewer. The large uncertainty is due to the relatively small sample size in
the observations (around 100 years), and thus is inherent to the statistical GEV approach.

251-252: "This indicates, incidentally, that the storyline approach is not providing larger effects
of climate change compared to the probabilistic approach, or possibly exaggerating these effect."
This sentence is confusing. Does this mean the storyline approach is 'not providing larger effects'

or is exaggerating effects (i.e. to make larger)? These seems to suggest storyline approach either
does't make the effects larger or it does make them larger. Please rephrase the sentence.

We agree with the reviewer that this sentence is confusing. Our intent was to say that based on
this particular event, we did not see larger (climate change) effects based on the storyline
approach.

259: This is a good point about mal-adaptation. I think it is a shame that this only now appears in
the paper and wasn't raised in the introduction.

We change the introduction to reflect this point.

270: Change "similarly" to "similar".

Figure 2: Subplot a needs a grey line in the legend to explain what the grey lines are, and caption
could explain how they relate to the blue line. You said (d) shows difference between (c) and (d),
think you meant (b) and (c).

Figure 3: This figure needs a lot more explanation in the caption. Possibly, this could be done in
part in the paragraph on lines 239-260 where the figure is discussed. Instead of simply making a
statement and referencing (Fig.3), instead reference (Fig.3a red line). This would make it easier
for the reader to connect the point your are discussing with the specific feature in Figure 3.

Figure 6. This needs more explanation in the caption.

OK, we will do so (on all 4 points above).

References

Blackport, R. and Fyfe, J. C.: Climate models fail to capture strengthening wintertime North Atlantic jet and impacts on
Europe, Science Advances, 8, eabn3112, https://doi.org/10.1126/sciadv.abn3112, publisher: American Association
for the Advancement of Science, 2022.

Deser, C. and Phillips, A. S.: A range of outcomes: the combined effects of internal variability and anthropogenic
forcing on regional cli- mate trends over Europe, Nonlinear Processes in Geophysics, 30, 63–84,
https://doi.org/10.5194/npg-30-63-2023, publisher: Copernicus GmbH, 2023.

Deser, C., Terray, L., and Phillips, A. S.: Forced and Internal Components of Winter Air Temperature Trends over North
America during the past 50 Years: Mechanisms and Implications, Journal of Climate, 29, 2237–2258,
https://doi.org/10.1175/JCLI-D-15-0304.1, publisher: American Meteorological Society Section: Journal of Climate,
2016.

Faranda, D., Messori, G., Jezequel, A., Vrac, M., and Yiou, P.: Atmospheric circulation compounds anthropogenic warm-
ing and impacts of climate extremes in Europe, Proceedings of the National Academy of Sciences, 120, e2214525120,
https://doi.org/10.1073/pnas.2214525120, publisher: Proceedings of the National Academy of Sciences, 2023.

Smoliak, B. V., Wallace, J. M., Lin, P., and Fu, Q.: Dynamical Adjustment of the Northern Hemisphere Surface Air
Temperature Field: Methodology and Application to Observations, Journal of Climate, 28, 1613–1629,
https://doi.org/10.1175/JCLI-D-14-00111.1, publisher: American Meteorological Society Section: Journal of Climate,
2015.

---

## Author Comment (AC2)

**Reply to Review of "An extreme cold Central European winter such as 1963 is unlikely but still possible despite climate change" (Reviewer #2)**

The manuscript asks the question of how would the cold 1963 European winter look like if similar atmospheric conditions were to develop in the recent, warmer, climate. Furthermore, it asks whether events as cold as winter 1963 might still be possible today. For this purpose, a number of different methodologies recently develop in the context of extreme event attribution are adopted and put to this test case. They conclude that climate change has warmed an event like winter 1963 by about 1.5 degrees and an event as cold as winter 1963 could still happen today, though it this is unlikely.

I have very much appreciated the multi-method approach which is adopted in the manuscript. The overall finding that the different methods provide very consistent answers is an important result, since it helps confirming the validity of the proposed approaches in this, and other, applications. I think this is an important result, which might benefit being stressed a bit more. The overall conclusion that such cold winter conditions would be warmer today is certainly not unexpected, but a quantification is still useful. The manuscript is well written, and the figures, though packed with perhaps too much information, are nice and clear. Overall, I have very few remarks for this manuscript, which I fully recommend for publication.

We thank the reviewer for the positive evaluation of our study, and we provide a more detailed response to the issues raised below. We think that stressing a bit more, as suggested, the comparison and comparable results of the different method is a good idea for the revised manuscript.

General (minor) comments:

- There is some repetition of the description of the approaches between the "method" section, and the result section. This would be fine if the methods section were at the end of the paper, but given the structure of this journal, I think repetitions could be reduced.

Thank you for point. We will reduce these repetitions for a revised manuscript.

- The authors could consider summarising the results from the different methods in a table.

Indeed, a table would make sense. We aim to summarize the results in a Table for the revised manuscript.

Specific minor comments

Line 31: It is not clear there is a long cold tail in the temperature distribution from Fig 1a, though I agree the tail is clearly evident in Fig 3. Please clarify.

This is true. The large outlier in Fig. 1a may be an indication, but is not clearly evident. We refer in a revised manuscript to literature that discusses that winter temperature distribution in Central Europe have a long negative tail.

Line 95: remove parenthesis in the citation of Shepherd 2016

Line 106: "with the same spacing" is not clear. Please clarify.

Line 115: please specify the estimated percentile

Line 117-125: this text should be removed from here, since it discusses results and not a method.

OK, we will adjust/clarify (all 4 points above).

Line 126: it's not clear to me why 2.5 is called "unconditional". Wouldn't 2.4 be unconditional too? The exact methods differ, but it seems to be me that the main difference between 2.4 and 2.5 is that the first is based on climate model data, and the latter on reanalysis data.

We agree that the terminology is somewhat unspecific. We called 2.5 "Unconditional" because the approach in 2.5 follows exactly the World Weather Attribution statistical approach (Philip et al. 2020) (on observations or reanalysis data), which is called "Unconditional" in the extreme event attribution literature. But we agree that even 2.5 is in fact conditional on the occurrence of a tail event, which is conceptually similar to 2.4 in models (although for this specific application it is not identical, because 2.4 uses the CESM2 large ensemble, i.e. tail events across the ensemble, while 2.5 is based on all winters because of the sparse observations).

Line 131: please add mean after DJF.

OK.

Line 137: do you mean a linear function of GSAT? If not, what function?

Yes, clarified.

Line 155: "utilising importance sampling techniques" would require some more discussion on how this is implemented. It received considerable less space than the boosting methodology. For example: is the reshuffling performed daily, or over blocks of a given length to better preserve autocorrelation?

This remark is justified. It received less space, because it was used mainly in comparison with the boosting, and has been implemented and described in a recent publication focusing on this method (Cadiou et al., 2023).

Line 163: please remove continuous.

OK.

Line 175: what is the CESM2-ETH ensemble? Please clarify

The CESM2-ETH ensemble is a small large ensemble, comparable to the CESM2-LE. The boosting methodology requires bit-by-bit reproducibility (to get to the restart files before the

actual event that is to be boosted), and we can achieve this required bit-by-bit reproducibility with the CESM2-ETH ensemble.

Line 175-185: this text could be reduced, since discussed in the results section.

Line 185: If the discussion of the method is repeated in the results section, such as here, then please add a reference to the section in which that method was discussed.

OK (to both points above).

Line 186: If I get it right, the winter cold extremes such as 1963 have warmed less than the DJF mean temperatures. But don't we expect cold extremes to warm faster than the mean, due to, e.g. plank feedback and polar amplification? Could you provide some discussion of this unexpected result? Can it be a consequence of the impact of the mean circulation trend on the mean temperature trend? Or other processes are at play.

This is an interesting point – indeed we expect cold extremes to increase faster than the mean caused mainly by temperature feedbacks such as lapse-rate feedback, Planck feedback and surface albedo feedback (also causing Arctic amplification) (e.g. Pithan and Mauritsen, 2014). In Central Europe, winter cold temperature anomalies are typically advected from northern regions, hence the mechanisms that cause AA likely also apply here.

Here, we analyse cold extremes on a seasonal time scale, and we indeed find this amplification in the CESM2 model (increase of about 1.4°C per 1°C GMT change, which is also larger than the mean change over Central Europe). Because we study observations, however, it is not so clear whether amplification of cold seasonal extremes can be shown or not: The sample is very short, and the (large) number of 2.5°C in the mean refers to the change in the decade 2014-2023 vs. 1951-1980, which is to some extent driven by circulation (and thus may possibly reverse in the future). We clarify in the manuscript that the large number of 2.5°C does not contradict the generally expected larger increase in the cold tail.

Line 261: Please note that method 2.5 had already answered this question. To the extent that return levels of such amplitude anomaly are still associated to a return period in the present day climate, events colder than 1963 are still possible. Please discuss.

Good point, yes, they generally seem to be possible based on the GEV analysis in 2.5. However, please note that the GEV approach bears large uncertainties because it is based on a relatively small sample of about 100 years in observations – so several (and independent) methods are needed to test this assertion. This brings us to 3.3, where we will include the comment about the GEV analysis from 2.5.

Line 239: Following on the previous point, I don't understand why the CESM2-LE approach is "conditional on the 1963 atmospheric circulation". That approach is just looking at the trend in a low percentile. It includes both changes in the magnitude and frequency of cold events. If the we assume that looking at a low percentile implies conditioning on "tail events", then the GEV approach from WWA should be equally considered a conditional analysis.

By that statement, we simply mean that the estimated warming of +1.4 to +1.6°C (independent of the approach) would yield a 1963-like winter of about -4.5° to -4.7°C below the seasonal average – which would still be very cold, but would only be possible if the 1963 circulation would return.

Line l 268-273: some redundant text, since the boosting is already discussed in the methods section.

OK.

L 300: The only information about the ability of CESM2 comes from Fig 4a, which is qualitative. Could you please quantify the mean temperature bias, and the bias in a low temperature quantile? That would be useful to add some confidence on the model results.

We will quantify these biases for a revised submission, along with the difference in the variance of seasonal temperature anomalies, which may be more relevant than the mean bias since we are working with anomalies.

Line 310: why not showing a map for the temperature anomaly associated to the empirical importance sampling? It could be showed in place of the second boosted CESM simulation which does not add much information.

We believe this is a good suggestion.

Fig 1 caption: add (blue dashed) after "atmospheric circulation" and (black dashed) after "global mean temperature".

Fig 2 caption: Please specify the method used to estimate the blu line in Fig 2a. b): replace over Germany with over Europe. The last (d) should be a (b).

Fig 3 caption: The main message seems that the different methods give consistent answer, more than there is uncertainty.

Fig 4: what is the unit of the Circulation (SLP) axis? Please specify.

Thanks for these points, we will clarify and expand the figure labels, also in response to reviewer #1.

References

Cadiou, C. and Yiou, P.: Simulating record-shattering cold winters of the 21st century in France, https://hal.science/hal-03900209, 2022.

Philip, S., Kew, S., van Oldenborgh, G.J., Otto, F., Vautard, R., van Der Wiel, K., King, A., Lott, F., Arrighi, J., Singh, R. and van Aalst, M., 2020. A protocol for probabilistic extreme event attribution analyses. Advances in Statistical Climatology, Meteorology and Oceanography, 6(2), pp.177-203.

Pithan, F. and Mauritsen, T., 2014. Arctic amplification dominated by temperature feedbacks in contemporary climate models. Nature geoscience, 7(3), pp.181-184.

---

## Author Comment (AC3)

**Reply to Reviewer #3**

This is a clear and concise paper, using a range of appropriate methodologies, to examine an important and useful question. I appreciate the discussion of the potential role of natural variability in the dynamical trends, and find the results and conclusions to be clear and well communicated. I recommend publication, subject to some minor comments below.

We thank the reviewer for the positive evaluation of our study, and we provide a more detailed response to the issues raised below.

Line 26. I suspect the c in O'conner should be capitalized? (also line 37 and other references to O'Conner (1963)

Yes, thanks.

Line 31. The long cold tail isn't immediately apparent from Fig. 1, I realized after looking at Fig. 3 that you do have the distribution shown on the far right, but this isn't referenced in the caption or labelled in any way, and so is easy to miss.

We will clarify, thanks (Reviewer #1 made a similar comment).

Line 80. Why SLP for the ERA5 model and z500 for the CESM LENS trained model? Is daily SLP not available in the CESM LENS? Do you think this influences the effectiveness of explaining surface temperature extremes?

There is no particular reason except to use to different published methods, based on SLP (Sippel et al., 2019) and based on z500 (Singh et al., 2023). Both methods show qualitatively similar behaviour, including the comparison of trends, with a somewhat slightly smaller explanatory power for the CESM-trained method. This is to be expected because of the dataset shift from training to application. We clarify in the respective section.

Line 95. Citation shouldn't be in parentheses

Thanks.

Line 130. I assume you mean 90-day running means?

Yes, fixed.

Line 174. CESM2-ETH is only defined later, on line 275.

Thanks, fixed.

Line 180. I think you mean Fig. 6 here? Also, it's a little confusing to reference Fig. 5 (or 6) before Fig. 4, although I understand that it's just to give some more information on the methodology.

We agree and will remove the early mention of Fig. 6 here.

Section 3.1: You show that average winter temperatures over Germany have increased by 2.5C, and explain that some of this is due to dynamical impacts, but it would be useful to have a quantitative assessment of how much thermodynamical warming has occurred over Germany

over this time period. In section 3.2 you show that the coldest winter (1963) would be only 1.4C warmer in the present climate, but also make the argument that the coldest winters occur when there is advection from regions that are experiencing greater thermodynamical warming. This would suggest to me that we might expect the coldest extremes to warm faster than the average temperature, but it is hard to make this comparison without know the thermodynamical contribution to average Germany winter temperatures.

This is a very important point. Reviewer #2 made a similar point, and we refer to the longer discussion in this reply to R2: Yes, we expect cold extremes to increase faster than the mean. We see this in the CESM2 model analysis, but it is difficult to show this concretely in observations because of the relatively short sample size and because we study seasonal extremes (rather than the coldest day per year or so). The 2.5°C increase is a large number (and is the total multidecadal change) and heavily influenced by the dynamical trend during this multidecadal period. We will clarify and discuss this point in a revised manuscript.

Sections 2 and 3: There could be clearer separation of the results between different sections. For example, I think section 2.4 reports results that might be (more) useful in section 3. Similarly, in line 245 you state that the winter 1963 event has a return period of 371 years in 2021, but then in the next section, 3.3, ask whether such an event as the 1963 winter would be possible in today's climate – it seems like you already answered this in the previous section if it has a return period of 371 years. I understand that you look at other methodologies in Section 3.3, but I suggest some re-arrangement to help this flow a little better.

Indeed the point about the GEV analysis around line 245 is important. We will clarify and rearrange such that it will become more clear that the section 3.3 is about the worst-case sampling methods (rather than the GEV fit), but we will mention that the GEV fit already indicates that 1963 would be possible (but unlikely) today.

Line 280. I don't think the word 'bias' is quite correct here. I would recommend 'residual' or similar.

We mean bias in the sense of the statistical model being able (or not) to explain the output of the climate model. But indeed this word may lead to confusion, so "residual" may be a better word here.

Fig. 4. The x-axis labels of circulation and albedo for the top row are closer to the plots below, and so look more like titles for those plots than x-axis labels for the upper plots.

Will be fixed.

Section 4. I would consider renaming this section as Discussion and Conclusions, as the first paragraph seems more discussion than conclusion.

Section 3 is already called "Results and Discussion". To account for the comment, we plan to rearrange such that the text related to discussion will end up in the "Results and Discussion" section.

References

Singh, J., Sippel, S., and Fischer, E.: Circulation dampened heat extremes intensification over the Midwest US and amplified over Western Europe, https://doi.org/10.21203/rs.3.rs-3094989/v1, 2023.

Sippel, S., Meinshausen, N., Merrifield, A., Lehner, F., Pendergrass, A. G., Fischer, E., and Knutti, R.: Uncovering the Forced Climate Response from a Single Ensemble Member Using Statistical Learning, Journal of Climate, 32, 5677–5699, https://doi.org/10.1175/JCLI- D-18-0882.1, publisher: American Meteorological Society Section: Journal of Climate, 2019.

Sippel, S., Meinshausen, N., Merrifield, A., Lehner, F., Pendergrass, A. G., Fischer, E., and Knutti, R.: Uncovering the Forced Climate Response from a Single Ensemble Member Using Statistical Learning, Journal of Climate, 32, 5677–5699, https://doi.org/10.1175/JCLI- D-18-0882.1, publisher: American Meteorological Society Section: Journal of Climate, 2019.

---

## Author Response (AR1)

**Reply to the Editor and all reviewers on egusphere-2023-2523, doi:10.5194/egusphere-2023-2523 (Sippel et al.)**

We thank the reviewers and the Editor for the careful and positive evaluation and handling of our manuscript. We provide detailed responses to all comments and questions in this document. In the following the reviewer comments appear in black with the author responses in red.

**Reply to Anonymous Referee #1 comment on egusphere-2023-2523, doi:10.5194/egusphere-2023-2523 (Sippel et al.)**

This study has investigated the likelihood of a re-occurrence of the extremely cold European winter of 1963 under present climate conditions and what such a winter would look like. To investigate this, the authors employ a range of techniques, some involve accounting for the dynamical features which gave rise to the extremely cold winter, while others are standard statistical methods, e.g. extreme value analysis. All the methods gave approximately similar answers, that such an event could occur today but is less likely, and if it did occur, the temperatures would be approximately 1.5 degrees warmer than the original 1963 event.

The paper provides a clear and thorough assessment of the theoretical occurrence of the 1963 winter under today's climate. It is an interesting study, and would have been more interesting if they had found that the winter could not occur today. However, since the conclusion is that an extremely cold winter from the past is less likely to occur and would be warmer if it did occur under today's warmer climate is not especially groundbreaking. The paper does raise the issue of mal-adaption by society towards warmer winters, and I think this is very valuable. I also think the comparison of the methods will be of interest to the community. Overall, I would recommend the paper be published with minor revisions.

We thank the reviewer for the positive evaluation of our study, and we provide a more detailed response to the issues raised below.

General Issues

1.  While the result that the cold winter can still occur and would be warmer is not wholly surprising, the paper does makes a good point that society may be adapting under the assumption of warmer winters while extremely cold winter are still very possible. I think this could be raised up in the paper. Perhaps introducing the idea as part of the motivation for the work so it is in the reader's minds as they go through.

We now provide a more thorough discussion of the impacts of extreme cold winter temperatures in Central Europe, and the importance of adaptation to extremes in this context. We focus on two sectors, health and energy, because both sectors are affected by multiple challenges in adaptation, namely ongoing societal transformations in many European countries (aging populations, transformation of the energy system), on top of the challenges associated with adapting to a changing climate. We argue that it is important to avoid the risk of maladaptation in a context where Europe has seen a sequence of mild winters for more than a decade, but where extreme cold temperatures remain a possibility (e.g., Quesada et al. 2022; Pinto et al., 2024). The text reads (l. 45-65): "Overall, the winter of 1963 serves as a canonical illustration of an extreme cold Central European winter. If a European winter of similar intensity as in 1963 were to re-occur in today's world, it would almost certainly have severe societal implications: Extreme cold winter temperatures cause adverse impacts in many sectors such as

health, transportation, infrastructure and energy (e.g., Pinto et al., 2024). As an example in the health sector, cold extremes are associated with increases in respiratory diseases affecting in particular the elderly and more vulnerable population (Curtis et al., 2017; Hajat and Haines, 2002), and increased mortality due to cardiovascular diseases (Curtis et al., 2017; Charlton-Perez et al., 2021). Vulnerable groups such as the temporary workforce, outdoor workers or the lower-income population with insufficient or precarious shelters, and insufficient access to energy or fuel, may be also strongly affected (Pinto et al., 2024). As another example, the energy sector is particularly affected by cold extremes. On the demand-side, there is a close relationship of temperature with demand for heating (Petrick et al., 2010; Zeniewski et al., 2023). On the supply side, several past events such as Texas in February 2021 have caused blackouts or near blackouts due to electric grid overload, or failure in sufficient power generation (Gruber et al., 2022). The combination of high demand and low supply may lead to associated price spikes from cold winter temperatures for electricity or natural gas, which may even lead to energy or fuel deprivation and mortality for vulnerable groups (Chirakijja et al., 2019). Given the vulnerability of different sectors to extreme cold temperatures in Europe and beyond, adaptation to those events, such as winterization of the energy sector (Zakeri et al., 2022) or cold wave preparedness and contingency measures in the health sector (Pinto et al., 2024), is crucial. The adaptation question may be particularly important in the context of ongoing societal transformations in Europe, such as an ageing population and the transformation of the energy system with a larger share of renewables in many European countries. Despite the strongly warming European winter climate with a sequence of mild winters for more than a decade in Central Europe, extreme cold temperatures can still occur (Quesada et al., 2023)(Pinto et al., 2024). Hence, potential extreme cold temperatures, despite a warming average winter climate, must be considered in adaptation strategies to avoid the risk of maladaptation."

2. It is a potentially provocative finding that the unconditional statistical method of using a fitted GEV gave as good a result as methods which incorporated knowledge of the dynamics. Many statistical based studies are criticized for not including knowledge of the dynamics of a situation, and here we have a case where this knowledge provided no additional benefit, and for an extreme event no less. Obviously, this is a single case and it could be random chance that the statistical method did so well. Given that the authors have used so many different methods, I think it would be a nice addition to the paper for the authors to briefly comment on how the methods compare, especially the unconditional statistical method with the methods incorporating knowledge of the dynamics.

We agree with the reviewer that the debate on different conditional and unconditional approaches to event attribution is relevant in this context. Indeed, the overall similar results in our study may occur by chance. But it may be noted nonetheless: The statistical method is fitted based on a long 1900-2023 period, with thus probably a relatively small dynamical trend over this long time scale (hence the thermodynamics likely dominate the unconditional analysis), which may partly explain the overall similar results). Interestingly, if the statistical method would be fitted on the shorter yet still quite long 1950-2023 period, the warming estimated obtained for a 1963-like winter would be much larger (best estimate of +2°C, instead of the +1.6°C from the long unconditional fit, or a similar result from a shorter fit but when dynamics are considered), because of the dynamical trend on this decadal time scale. We have included a short discussion in the revised manuscript:

"A statistical fit based on the shorter (yet still quite long) post-1950 period, would yield a larger estimate of winter temperature change between 1963 and today (best estimate of +2.0°C, rather

than +1.6°C when based on the longer period). This is likely because the post-1950 circulation-induced trend is congruent with thermodynamical warming, and the shorter period estimate would thus implicitly assume that the total winter temperature trend over Germany, including the circulation effects, is forced. Hence the circulation trends would implicitly be assumed as forced effects through the global temperature covariate. This illustrates the potential pitfalls of using relatively short time series in an unconditional manner." (l. 278-284)

In addition, we have tested the robustness of the long-period GEV fit against different choices of block length (using the long DWD time series, for blocks of 1-10 years, using GISTEMP GMST as covariate). Time-series were constructed such that they ended on a complete block, with hence partly varying start dates but all earlier than 1889.  Figure 1 below shows the results for key parameters: mu0 (stationary mean temperature), alpha (linear trend parameter), sigma0 (GEV scale parameter) and shape (GEV shape parameter), plus the estimated return period of the 1962-3 winter and the probability ratio (specifically, the change in likelihood between the 2021-2 and 1962-3 winters). The red bars indicate a bootstrapped 95% confidence interval for each parameter, and the dashed lines indicate the 95% interval for the original analysis; the dots and solid line are the corresponding best estimates. For reference we have also added a blue bar indicating the 95% confidence interval when all years from 1950 onwards are used. Apart from mu0 (which is strongly dependent on the block length - unsurprisingly, since taking n-year blocks effectively removes the warmest 1/n years from the dataset) the results are robust with increasing  uncertainties due to the well-known bias-variance trade-off). Importantly, for all block lengths the estimated linear trend falls within the 95% confidence interval estimated from all years, as do the return periods and most of the PRs. The shape parameter changes quite a bit but perhaps that is to be expected, given how much the centre of the distribution shifts with the block length.

Overall, the GEV results are far more sensitive to truncation of the time series than to changing the block length; and the results of the unconditional analysis are robust to the choice of GEV block length.

[Figure]

*Figure 1:* Influence of block length on GEV results (all details described in the text).

3. The title is a little sensationalist and I'd suggest changing it to match the style of the WCD journal.

Thank you for this feedback. We have changed the title to directly reflect our research question: "Could an extreme cold Central European winter such as 1963 happen again despite climate change?". We hope that this title matches with the style of the WCD journal.

4. The second paragraph in the Results section discusses the failure of models to show pronounced forced changes in atmospheric circulation. This is not a topic that is really investigated by this study. CESM2 simulations were run to perform the model boosting analysis, but there is no assessment of how or why or to what extent CESM2 fails to show forced changes. The discussion reads more like a commentary on the failure of models in this particular aspect, and does feel connected to the rest of the study. If this is a main motivation or theme in this study, then this failure of models should be introduced in the introduction and its implications on the use of model data in this study needs to be assessed, not merely commented on. Without expanding on the issue raised in this paragraph and incorporating it more fully into the study, I'd recommend removing the paragraph, more specifically, sentences from line 204 to 212.

Thank you for highlighting this important point. Indeed it is not the goal of this study to investigate the ability or failure of models with respect to simulating multi-decadal trends in atmospheric circulation. There are many other studies that look at this phenomenon, and we cite them (Blackport and Fyfe 2022, Faranda et al., 2023).

We nevertheless argue that this paragraph contains important contextual information, which informs the assumptions of our study, and we have rephrased the paragraph to make the reasoning more clear. The revised paragraph reads (l. 225-239): "Beyond thermodynamical effects, however, atmospheric circulation changes have contributed to warming winter temperatures over Central Europe, as evidenced by the circulation trend in Fig. 1a. The positive circulation contribution to warming, with less frequent cold spells, is consistent with other studies (Vautard et al., 2009; Faranda et al. 2023), and reflects more frequent zonal flows and less frequent blocked flows in the recent past (Blackport and Fyfe, 2022; Faranda et al., 2023). Model simulations, including CESM2, however, show little or no evidence for pronounced forced changes (Blackport and Fyfe, 2022). Therefore, the future of forced regional atmospheric circulation changes remains highly uncertain Shepherd, 2014; Zappa and Shepherd, 2017; Blackport and Fyfe, 2022; Faranda et al., 2023). It is beyond the scope of our study to investigate models' circulation trends, which have been analysed in several past studies (Blackport and Fyfe, 2022; Faranda et al., 2023). However, the discrepancy between model simulations and observations carries important implications for understanding and constraining the potential for future cold winters: If the circulation trend was indeed forced, but missed by climate models, it would be less likely to see winter 1963 circulation conditions again. If, however, the dynamical trend would be due to natural variability, circulation conditions similar to 1963 may appear again with equal probability. It is the latter scenario that would bring severe risks for European societies. Therefore, we will focus on a storyline approach, which has been conceived precisely for the conditions of large dynamical uncertainties (Shepherd, 2016). In this context of dynamical uncertainty, we analyse in the next subsection how winter 1963 circulation conditions would play out in a present-day climate."

Note that we mention the CESM2-specific circulation-induced trend in the Method section that first introduces CESM2 (but not in the paragraph mentioned by the reviewer, to keep the reasoning generic): "The CESM2 model is used as it is shown to perform well for European regional climate (Deser and Phillips, 2023), and indeed the distributions between CESM2-LE and ERA5 compare well visually (Fig.3a). The standard deviation of the detrended 1950-2023 winter seasonal temperature distribution is 1.40°C, which is slightly smaller than the winter temperature standard deviation in ERA5 (1.68°C). We estimate intensity changes, but we do not derive probabilistic return period or frequency estimates from the model. The model shows only a very small ensemble mean circulation-induced trend over the historical period (Fig. 10c in Deser and Phillips (2023))." (l. 137-143)

5. Figure captions for figures 2, 3, and 6 need to be expanded to better explain what is being shown in the figures. This is especially true of figures 3 which is important for the paper.

We have expanded and clarified the figure captions of Figures 2, 3, and 6. We have also modified Figure 3 slightly to ease the interpretation.

Minor Issues

24: You use the expression "led to". This suggests some precession in time, that the pressure anomalies occurred and then afterwards, there was a negative NAO anomaly. Perhaps change to "resulted in" or "comprised".

Correct, we did not mean precession in time, but indeed "resulted in" or "associated with". Fixed.

65-90: The dynamical adjustment method. Is it reasonable to separate the dynamic and thermodynamic influences on surface temperature in such a linear way? Consider adding a sentence or two discussing the caveats or limitations of this assumption/approach.

Important point. In the dynamical adjustment literature, this separation in both components appears quite standard (Smoliak et al., 2015, Deser et al., 2016, Deser et al., 2023). But of course it needs to be acknowledged that this is an assumption that may bear some caveats for the interpretation: "These approaches assume a linear separation between both components, which is a limitation of these techniques. " (l. 90)

78: "The first dynamical adjustment approach (dark blue line in Fig. 1) uses ERA5 to train the regression model, and the spatial pattern of sea level pressure (SLP) over a circulation domain over Europe and the North Atlantic." Possibly it is just my reading of the sentence, but it feels a little awkward. The 'and' feels like it is part of the regression statement. Consider changing this to "along with".

Done.

82: Start a new paragraph at "We use a second method…" It provides a cleaning break when reading. Possibly change to "We also use a second…"

Thanks, done.

169-183: Please revise the structure of this paragraph. It jumps straight into what the method leads to and only describes the method itself towards the end of the paragraph. I suspect this is one of those cases where the author is so familiar with the method, he forgets that the reader may not understand what he is talking about from the beginning.

We have restructured the paragraph, such that the main idea is explained first, followed by the more technical details of the implementation.

174-181: Please explain in the paragraph why you use the single coldest winter in December for the first boosting, but two coldest Januarys for the second boosting.

Our aim was to test (in the context of a "storyline") whether a winter of the 2020's could be even as cold as the observed 1963 winter. Therefore we started with the 1st to 15th December of the coldest December in the 2020's (there was a colder December in the 2000's which we did not boost). We then realized that most ensemble members of this coldest December in the 2020's were going back to the climatologically expected "normal" range in the following January, but there were two ensemble members that showed exceptionally cold (but almost identically cold) conditions in the following January. Because resources allowed it, we decided to boost both of these events, which yield relatively similar "coldest DJF of the 2020's" conditions, and are now both shown in Fig. 6.  We clarified: "The two coldest simulations were selected, because computational resources allowed it." (l. 206-207)

176: Please write the dates out in full. Using "01.12 to 15.12" could be written as "1st to 15th December".

180: Again, write the dates out in full.

219: Remove the double brackets.

OK, thanks, all three above fixed.

245: The return period of the 1963 event was 119 years. For such an event to occur today, the return period would be 371 years. However, the uncertainty for the occurrence today is 97 to 7680 years. That is to say, the return period of the event today may still be 119 years at the 95% level. This should be commented on.

We agree with the reviewer: The large uncertainty is due to the relatively small sample size in the observations (around 100 years), and thus is inherent to the statistical GEV approach. But if the statistical fit would be biased such that the true value today would be 119 years, then the event would also have had a lower return period in 1963 (because the bias would be in the same direction).

251-252: "This indicates, incidentally, that the storyline approach is not providing larger effects of climate change compared to the probabilistic approach, or possibly exaggerating these effect." This sentence is confusing. Does this mean the storyline approach is 'not providing larger effects' or is exaggerating effects (i.e. to make larger)? These seems to suggest storyline approach either does't make the effects larger or it does make them larger. Please rephrase the sentence.

We have simplified the wording to avoid the ambiguity which arose from the construction of the sentence.

259: This is a good point about mal-adaptation. I think it is a shame that this only now appears in the paper and wasn't raised in the introduction.

We revised the introduction carefully, such that it now discusses the issue of cold winter impacts, and potential maladaptation, more extensively.

270: Change "similarly" to "similar".

done

Figure 2: Subplot a needs a grey line in the legend to explain what the grey lines are, and caption could explain how they relate to the blue line. You said (d) shows difference between (c) and (d), think you meant (b) and (c).

fixed, thanks.

Figure 3: This figure needs a lot more explanation in the caption. Possibly, this could be done in part in the paragraph on lines 239-260 where the figure is discussed. Instead of simply making a statement and referencing (Fig.3), instead reference (Fig.3a red line). This would make it easier for the reader to connect the point your are discussing with the specific feature in Figure 3.

We have expanded the caption of Fig. 3 substantially. We have indicated (a) or (b) in the text that discusses the figure, and we have revised the figure itself such that it is now less busy and easier to read.

Figure 6. This needs more explanation in the caption.

We have expanded the caption of Fig. 6, also in line with a comment by Reviewer 2.

**Reply to Review of "An extreme cold Central European winter such as 1963 is unlikely but still possible despite climate change" (Reviewer #2)**

The manuscript asks the question of how would the cold 1963 European winter look like if similar atmospheric conditions were to develop in the recent, warmer, climate. Furthermore, it asks whether events as cold as winter 1963 might still be possible today. For this purpose, a number of different methodologies recently develop in the context of extreme event attribution are adopted and put to this test case. They conclude that climate change has warmed an event like winter 1963 by about 1.5 degrees and an event as cold as winter 1963 could still happen today, though it this is unlikely.

I have very much appreciated the multi-method approach which is adopted in the manuscript. The overall finding that the different methods provide very consistent answers is an important result, since it helps confirming the validity of the proposed approaches in this, and other, applications. I think this is an important result, which might benefit being stressed a bit more. The overall conclusion that such cold winter conditions would be warmer today is certainly not unexpected, but a quantification is still useful. The manuscript is well written, and the figures, though packed with perhaps too much information, are nice and clear. Overall, I have very few remarks for this manuscript, which I fully recommend for publication.

We thank the reviewer for the positive evaluation of our study, and we provide a more detailed response to the issues raised below.

We think that stressing a bit more, as suggested, the comparison and comparable results of the different methods is a good idea and we included this in the revised manuscript in the Results&Discussion section (Subsection 3.2, line ~285-289), and in the conclusion. Please note that we have also added more discussion on the fact that short GEV fits, e.g. here based only on the 1950-2023 period, don't agree with the other methods, because of the positive circulation trend over this shorter sub-period.

General (minor) comments:

- There is some repetition of the description of the approaches between the "method" section, and the result section. This would be fine if the methods section were at the end of the paper, but given the structure of this journal, I think repetitions could be reduced.

Thank you for flagging these repetitions. We have removed several repetitions that are similar to the description in the Methods part in Results&Discussion, especially in Subsection 3.1 on dynamical adjustment, and in Section 3.2 on the circulation analogue approach. In addition, we removed from the methods section some text that previously discussed CESM2 amplification results and that actually belongs to Results&Discussion (see also your comment below).

- The authors could consider summarising the results from the different methods in a table.

Thank you for this suggestion. We have indeed created a table (see Table 1), which summarizes the attribution statements.

Specific minor comments

Line 31: It is not clear there is a long cold tail in the temperature distribution from Fig 1a, though I agree the tail is clearly evident in Fig 3. Please clarify.

It is true that the long cold tail is not clearly visible in Fig. 1a, where we did not compute a probability distribution. But we confirm that for the ERA5 data (black lines/points) it is the exact same data shown in Fig. 1a and Fig. 3.

The long cold tail in the winter temperature distribution in Central Europe can indeed be understood from a synoptic meteorology perspective (line 31-35): "The long cold tail of the observed distribution in Fig. 1a due to winter 1963 is not an artefact of the short observed record, but can be well explained by the prevailing circulation patterns (Loikith and Neelin 2019): under normal winter conditions Central Europe is under westerly flow, while the climatologically coldest air resides to the East. Rare, anomalous reversals towards easterly flow thus create the observed long, non-Gaussian cold tail (Loikith and Neelin 2019), which is further enhanced by snow-albedo feedbacks (Groisman et al. 1994)."

Line 95: remove parenthesis in the citation of Shepherd 2016

done

Line 106: "with the same spacing" is not clear. Please clarify.

Clarified: "which are apart from each other by at least six days"

Line 115: please specify the estimated percentile

done

Line 117-125: this text should be removed from here, since it discusses results and not a method.

Indeed, this is a clear part of results, which has erroneously appeared in the methods section. We have integrated this part with the respective results section; and hence the repetition is removed (see also your comment above).

Line 126: it's not clear to me why 2.5 is called "unconditional". Wouldn't 2.4 be unconditional too? The exact methods differ, but it seems to be me that the main difference between 2.4 and 2.5 is that the first is based on climate model data, and the latter on reanalysis data.

We agree that the terminology is somewhat unspecific. We called 2.5 "Unconditional" because the approach in 2.5 follows exactly the World Weather Attribution statistical approach (Philip et al. 2020) (on observations or reanalysis data), which is called "Unconditional" in the extreme event attribution literature. But we agree that even 2.5 is in fact conditional on the occurrence of a tail event, which is conceptually similar to 2.4 in models (although for this specific application it is not identical, because 2.4 uses the CESM2 large ensemble, i.e. tail events across the

ensemble, while 2.5 is based on all winters because of the sparse observations). We have clarified this point in the text.

Line 131: please add mean after DJF.

done

Line 137: do you mean a linear function of GSAT? If not, what function?

Yes, clarified.

Line 155: "utilising importance sampling techniques" would require some more discussion on how this is implemented. It received considerable less space than the boosting methodology. For example: is the reshuffling performed daily, or over blocks of a given length to better preserve autocorrelation?

This remark is justified. It received less space, because it was used mainly in comparison with the boosting, and has been implemented and described in a recent publication focusing on this method (Cadiou and Yiou, 2024). We have expanded the description of the importance sampling, to make sure the key idea and technical steps are described thoroughly, including the fact that the reshuffling is conducted on a daily time scale, and we refer to Cadiou and Yiou(2024) for technical details. In addition, we modified Fig. 5 such that it also shows the corresponding maps obtained from the empirical importance sampling in comparison to the boosting technique.

Line 163: please remove continuous.

OK.

Line 175: what is the CESM2-ETH ensemble? Please clarify

The CESM2-ETH ensemble is the 30-year ensemble that we use for selecting candidates for model boosting. The boosting methodology requires bit-by-bit reproducibility (to get to the restart files before the actual event that is to be boosted), and we can achieve this required bit-by-bit reproducibility with the CESM2-ETH ensemble. We have clarified the text accordingly.

Line 175-185: this text could be reduced, since discussed in the results section.

We have significantly shortened this text, in particular by omitting results/interpretation that are then indeed discussed in the result section. We also removed in the corresponding Result section the parts that are in fact already described in Methods.

Line 185: If the discussion of the method is repeated in the results section, such as here, then please add a reference to the section in which that method was discussed.

We have removed method discussions in the Results section.

Line 186: If I get it right, the winter cold extremes such as 1963 have warmed less than the DJF mean temperatures. But don't we expect cold extremes to warm faster than the mean, due to, e.g. plank feedback and polar amplification? Could you provide some discussion of this unexpected result? Can it be a consequence of the impact of the mean circulation trend on the mean temperature trend? Or other processes are at play.

This is an interesting point – indeed we expect cold extremes to increase faster than the mean caused mainly by temperature feedbacks such as lapse-rate feedback, Planck feedback and surface albedo feedback (also causing Arctic amplification) (e.g. Pithan and Mauritsen, 2014). In Central Europe, winter cold temperature anomalies are typically advected from northern regions, hence the mechanisms that cause AA likely also apply here. Here, we analyse cold extremes on a seasonal time scale, and we indeed find this amplification in the CESM2 model (increase of about 1.4°C per 1°C GMT change, which is also larger than the mean change over Central Europe). Because we study observations, however, it is not so clear whether amplification of cold seasonal extremes can be shown or not: The sample is very short, and the (large) number of 2.5°C in the mean refers to the  change in the decade 2014-2023 vs. 1951-1980, which is to some extent driven by circulation (and thus may possibly reverse in the future).

We clarify in the manuscript that more extreme cold events are indeed expected to warm faster than the mean, and that we see this in the model simulations (l. 267-271): "We acknowledge that more extreme cold events than the '119-year event' studied above would potentially show even a higher amplification (Tamarin-Brodsky et al., 2020). This is because a weaker meridional temperature gradient, and large warming over Arctic regions (Pithan and Mauritsen, 2014) implies particularly strong warming of cold extremes in Central Europe that are caused partly by advection from those regions (Tamarin-Brodsky et al., 2020)." and later in l.338-341: "It is particularly remarkable that the model is capable of simulating such cold conditions in the 2020's, as the cold tail of the winter temperature distribution tends to warm faster than average conditions due to well-understood physical reasons (Tamarin-Brodsky et al., 2020). This is consistent with the analysis of the CESM2 winter amplification in Fig. 3b."

Line 261: Please note that method 2.5 had already answered this question. To the extent that return levels of such amplitude anomaly are still associated to a return period in the present day climate, events colder than 1963 are still possible. Please discuss.

Good point, yes, they generally seem to be possible based on the GEV analysis in 2.5. However, please note that the GEV approach bears large uncertainties because it is based on a relatively small sample of about 100 years in observations – so several (and independent) methods are needed to test this assertion. This brings us to 3.3, where we included a brief comment about the GEV analysis. "Based on the statistical GEV analysis presented in Subsection 3.2, cold temperatures such as in winter 1963 would still be possible today, albeit very unlikely (best estimate of a 371-year return event). However, the statistical analysis is based on the relatively short observational record, with large uncertainties, and here our goal is to develop storylines of such cold winter temperatures based on independent rare event sampling methods." (l. 297-300)

Line 239: Following on the previous point, I don't understand why the CESM2-LE approach is "conditional on the 1963 atmospheric circulation". That approach is just looking at the trend in a

low percentile. It includes both changes in the magnitude and frequency of cold events. If the we assume that looking at a low percentile implies conditioning on "tail events", then the GEV approach from WWA should be equally considered a conditional analysis.

You are correct, the CESM2-LE approach is not conditional on atmospheric circulation. We have moved the sentence to further above, where the circulation-conditional approach is discussed.

Line l 268-273: some redundant text, since the boosting is already discussed in the methods section.

Yes, we have integrated this text in the Methods section.

L 300: The only information about the ability of CESM2 comes from Fig 4a, which is qualitative. Could you please quantify the mean temperature bias, and the bias in a low temperature quantile? That would be useful to add some confidence on the model results.

Yes, important point. Note that we show the distribution of ERA5 (1950-2023) and the CESM2-LE (1950-2023) in the violin plots in Fig. 3 side-by-side, where they visually agree very well. Since the study is based on anomalies (with reference to the 1981-2010 period), we quantify the standard deviation of the (detrended) distributions and a low temperature quantile:

SD (ERA5): 1.68°C, SD (CESL2-LE): 1.40°C

ERA5, low temperature quantile (winter 1963): –6.3°C (48 up to 1102 year return event)

CESM2, low temperature quantile (48 up to 1102 year return event): -5.9°C up to -3.7°C

The visual and quantitative comparison shows that the simulated distribution by CESM2-LE agrees reasonably well with ERA5. CESM2 tends to underestimate the variability slightly (standard deviation of 1.4°C as opposed to 1.68°C in ERA5), which also translates to an underestimation of the intensity of cold quantiles in the model. However, importantly, we do not interpret the model simulations probabilistically (i.e., we don't calculate return period based on the model), and the fact that the model still simulates winter conditions colder than 1963 in the 2020s shows that the model is capable of simulating very cold winter conditions (though not with the same probabilistic frequency as in observations). We have amended the text (l. 137-140): "The CESM2 model is used as it is shown to perform well for European regional climate (Deser et al., 2023), and indeed the distributions between CESM2-LE and ERA5 compare favourably (Fig.3a). The standard deviation of the detrended 1950-2023 winter seasonal temperature distribution is 1.40°C, which is slightly smaller than the winter temperature standard deviation in ERA5 (1.68°C). We estimate intensity changes, but we do not derive probabilistic return period or frequency estimates from the model."

Line 310: why not showing a map for the temperature anomaly associated to the empirical importance sampling? It could be showed in place of the second boosted CESM simulation which does not add much information.

Thank you. This is a very good suggestion to highlight the empirical importance sampling results a bit more. We have implemented it in Fig. 5.

Fig 1 caption: add (blue dashed) after "atmospheric circulation" and (black dashed) after "global mean temperature".

done

Fig 2 caption: Please specify the method used to estimate the blue line in Fig 2a. b): replace over Germany with over Europe. The last (d) should be a (b).

Done, thanks.

Fig 3 caption: The main message seems that the different methods give consistent answer, more than there is uncertainty.

Correct, and we have changed the caption accordingly.

Fig 4: what is the unit of the Circulation (SLP) axis? Please specify.

Very important point: The axis specifies the prediction of temperature by using atmospheric circulation as predictor. So the axis label should read: "Predicted Daily Temperature Anomaly (°C) based on circulation predictors" (and so forth). Fixed.

**Reply to Reviewer #3**

This is a clear and concise paper, using a range of appropriate methodologies, to examine an important and useful question. I appreciate the discussion of the potential role of natural variability in the dynamical trends, and find the results and conclusions to be clear and well communicated. I recommend publication, subject to some minor comments below.

We thank the reviewer for the positive evaluation of our study, and we provide a more detailed response to the issues raised below.

Line 26. I suspect the c in O'conner should be capitalized? (also line 37 and other references to O'Conner (1963)

Yes, thanks.

Line 31. The long cold tail isn't immediately apparent from Fig. 1, I realized after looking at Fig. 3 that you do have the distribution shown on the far right, but this isn't referenced in the caption or labelled in any way, and so is easy to miss.

We have clarified this aspect in the caption of Fig. 3. In the text, where we also discuss the meteorological reasons for the cold tail.

Line 80. Why SLP for the ERA5 model and z500 for the CESM LENS trained model? Is daily SLP not available in the CESM LENS? Do you think this influences the effectiveness of explaining surface temperature extremes?

There is no particular reason except to use two different published methods, based on SLP (Sippel et al., 2019) and based on z500 (Singh et al., 2023). Both methods show qualitatively similar behaviour, including the comparison of trends, with a somewhat slightly smaller explanatory power for the CESM-trained method. This is to be expected because of the dataset shift from training to application. We clarified in the respective section.

Line 95. Citation shouldn't be in parentheses

done

Line 130. I assume you mean 90-day running means?

Yes, fixed.

Line 174. CESM2-ETH is only defined later, on line 275.

Thanks, fixed (see reply to R2).

Line 180. I think you mean Fig. 6 here? Also, it's a little confusing to reference Fig. 5 (or 6) before Fig. 4, although I understand that it's just to give some more information on the methodology.

We have removed the early mentioning of Fig. 5/6.

Section 3.1: You show that average winter temperatures over Germany have increased by 2.5C, and explain that some of this is due to dynamical impacts, but it would be useful to have a quantitative assessment of how much thermodynamical warming has occurred over Germany over this time period. In section 3.2 you show that the coldest winter (1963) would be only 1.4C

warmer in the present climate, but also make the argument that the coldest winters occur when there is advection from regions that are experiencing greater thermodynamical warming. This would suggest to me that we might expect the coldest extremes to warm faster than the average temperature, but it is hard to make this comparison without know the thermodynamical contribution to average Germany winter temperatures.

This is a very important point. Reviewer #2 made a similar point, and we refer to the longer discussion in this reply to R2: Yes, we expect cold extremes to increase faster than the mean. We see this in the CESM2 model analysis, but it is difficult to show this concretely in observations because of the relatively short sample size and because we study seasonal extremes (rather than the coldest day per year or so). The 2.5°C increase is a large number, but it is heavily influenced by the dynamical trend over the last decades, and internal variability because it reflects only the last decade. We have clarified the discussion about the expected amplification of cold extremes, exceeding the mean change, in the revised manuscript; and we also discuss that the 2.5°C warming includes a large circulation-induced component that potentially may reflect internal variability (Subsection 3.1).

Sections 2 and 3: There could be clearer separation of the results between different sections. For example, I think section 2.4 reports results that might be (more) useful in section 3. Similarly, in line 245 you state that the winter 1963 event has a return period of 371 years in 2021, but then in the next section, 3.3, ask whether such an event as the 1963 winter would be possible in today's climate – it seems like you already answered this in the previous section if it has a return period of 371 years. I understand that you look at other methodologies in Section 3.3, but I suggest some re-arrangement to help this flow a little better.

Yes, these are two important points. We have thoroughly rearranged the text to separate Methods and Results more stringently, and avoid any duplications.  See all detailed changes in the response to Reviewer 2.

Regarding the GEV analysis and the research question of Subsection 3.3: Indeed, our goal was here to develop storylines using rare event sampling methods – rather than GEV analysis. But it is of course true that the GEV analysis would also provide an answer to this question. Hence, we have added text at the beginning of Subsection 3.3 to make this clear:  "Based on the statistical GEV analysis presented in Subsection 3.2, cold temperatures such as in winter 1963 would still be possible today, albeit very unlikely (best estimate of a 371-year return event). However, the statistical analysis is based on the relatively short observational record, with large uncertainties, and here our goal is to develop storylines of such cold winter temperatures based on independent rare event sampling methods." (l. 296-299)

Line 280. I don't think the word 'bias' is quite correct here. I would recommend 'residual' or similar.

We replaced the word bias by the word "unexplained residual" – which indeed captures better what we mean here.

Fig. 4. The x-axis labels of circulation and albedo for the top row are closer to the plots below, and so look more like titles for those plots than x-axis labels for the upper plots.

Thanks, fixed.

Section 4. I would consider renaming this section as Discussion and Conclusions, as the first paragraph seems more discussion than conclusion.

We have moved this paragraph to the Results & Discussion section, as it indeed reflect a discussion rather than conclusion.

**References**

Blackport, R. and Fyfe, J. C.: Climate models fail to capture strengthening wintertime North Atlantic jet and impacts on Europe, Science Advances, 8, eabn3112, https://doi.org/10.1126/sciadv.abn3112, publisher: American Association for the Advancement of Science, 2022.

Cadiou, C. and Yiou, P.: Simulating record-shattering cold winters of the beginning of the 21st century in France, EGUsphere [preprint], https://doi.org/10.5194/egusphere-2024-612, 2024

Deser, C. and Phillips, A. S.: A range of outcomes: the combined effects of internal variability and anthropogenic forcing on regional cli- mate trends over Europe, Nonlinear Processes in Geophysics, 30, 63–84, https://doi.org/10.5194/npg-30-63-2023, publisher: Copernicus GmbH, 2023.

Deser, C., Terray, L., and Phillips, A. S.: Forced and Internal Components of Winter Air Temperature Trends over North America during the past 50 Years: Mechanisms and Implications, Journal of Climate, 29, 2237–2258, https://doi.org/10.1175/JCLI-D-15-0304.1, publisher: American Meteorological Society Section: Journal of Climate, 2016.

Faranda, D., Messori, G., Jezequel, A., Vrac, M., and Yiou, P.: Atmospheric circulation compounds anthropogenic warm- ing and impacts of climate extremes in Europe, Proceedings of the National Academy of Sciences, 120, e2214525120, https://doi.org/10.1073/pnas.2214525120, publisher: Proceedings of the National Academy of Sciences, 2023.

Smoliak, B. V., Wallace, J. M., Lin, P., and Fu, Q.: Dynamical Adjustment of the Northern Hemisphere Surface Air Temperature Field: Methodology and Application to Observations, Journal of Climate, 28, 1613–1629, https://doi.org/10.1175/JCLI-D-14-00111.1, publisher: American Meteorological Society Section: Journal of Climate, 2015.

Singh, J., Sippel, S., and Fischer, E.: Circulation dampened heat extremes intensification over the Midwest US and amplified over Western Europe, https://doi.org/10.21203/rs.3.rs-3094989/v1, 2023.

Sippel, S., Meinshausen, N., Merrifield, A., Lehner, F., Pendergrass, A. G., Fischer, E., and Knutti, R.: Uncovering the Forced Climate Response from a Single Ensemble Member Using Statistical Learning, Journal of Climate, 32, 5677–5699, https://doi.org/10.1175/JCLI- D-18-0882.1, publisher: American Meteorological Society Section: Journal of Climate, 2019.

Philip, S., Kew, S., van Oldenborgh, G.J., Otto, F., Vautard, R., van Der Wiel, K., King, A., Lott, F., Arrighi, J., Singh, R. and van Aalst, M., 2020. A protocol for probabilistic extreme event attribution analyses. Advances in Statistical Climatology, Meteorology and Oceanography, 6(2), pp.177-203.
Pithan, F. and Mauritsen, T., 2014. Arctic amplification dominated by temperature feedbacks in contemporary climate models. Nature geoscience, 7(3), pp.181-184.

---

## Author Response (AR2)

**Reply to the Editor and all reviewers on egusphere-2023-2523, doi:10.5194/egusphere-2023-2523 (Sippel et al.)**

We thank the reviewers and the Editor for the careful and positive evaluation of our manuscript.

Referee#1:

"As noted previously this study has investigated the likelihood of a reoccurrence of the extremely cold European winter of 1963 under present climate conditions and what such a winter would look like using a range of techniques, some accounting for the dynamical features which gave rise to the extremely cold winter while others are statistical in nature. On further consideration, I note that surface temperature variability is expected to decrease under global warming which, all other things being equal, would also contribute to cold Eurasian winters becoming less frequent. There was a nice perspective on this shifting of probabilities in a recent synthesis paper by Outten et al. 2022. On the issue of decreasing surface temperature variability, I realised the Blackport and Kushner 2016 paper was not mentioned, although there is reference to the Schneider et al. 2015 and Holmes et al. 2016 papers, so perhaps it is not needed.

Overall however, the authors have been extremely thorough in responding to all the general and specific comments that I raised in my first review of their work. I do feel that the paper has been improved by their efforts, and I hope they concur with this assessment. The references I mention above are only raised as suggestions to the authors for the sake of completeness but I am delighted to recommend the paper for publication, with or without the addition of these two reference.

Outten, S., Li, C., King, M. P., Suo, L., Siew, P. Y. F., Cheung, H., Davy, R., Dunn-Sigouin, E., Furevik, T., He, S., Madonna, E., Sobolowski, S., Spengler, T., and Woollings, T.: Reconciling conflicting evidence for the cause of the observed early 21st century Eurasian cooling, Weather Clim. Dynam., 4, 95–114, https://doi.org/10.5194/wcd-4-95-2023, 2023.

Blackport, R. and Kushner, P. J.: The transient and equilibrium climate response to rapid summertime sea ice loss in CCSM4, J. Climate, 29, 401–417, https://doi.org/10.1175/JCLI-D-15-0284.1, 2016."

Thanks for these references, which are indeed very relevant. We have added both references to our discussion of Arctic influences on mid-latitude wintertime climate, around l. 224 and l. 230.

Referee#2:

"I thank the authors for fully addressing my previous comments and concerns. I think the manuscript is now very clear and it represents a valuable contribution to the topic of extreme events attribution. Hence, I recommend this manuscript for publication in its current form."

Thank you for the positive evaluation.